



**Measurement report: Intensive biomass burning emissions and**
**rapid nitrate formation drive severe haze formation in Sichuan**
**basin, China: insights from aerosol mass spectrometry**
Zhier Bao[1], Xinyi Zhang[1], Qing Li[1], Jiawei Zhou[1], Guangming Shi[2], Li Zhou[2], Fumo Yang[2],
Shaodong Xie[3], Dan Zhang[4], Chongzhi Zhai[4], Zhenliang Li[4], Chao Peng[4], and Yang Chen[1]
[1] Chongqing Institute of Green and Intelligent Technology, Chinese Academy of Sciences,
Chongqing, 400714, China
[2] Department of Environmental Science and Engineering, College of Architecture and Environment,
Sichuan University, Chengdu 610065, China
[3] SKL-ESPC and BIC-ESAT, College of Environmental Sciences and Engineering, Peking
University, Beijing 100871, China
[4] Academy of Environmental Science, Chongqing, 401147, China
Correspondence: Yang Chen (chenyang@cigit.ac.cn)
**Abstract**
Haze pollution is a severe environmental problem, caused by elevation of fine particles
(aerodynamic diameter < 2.5 μm, $PM_{2.5}$), which is related to secondary aerosol formation,
unfavourable synoptic conditions, regional transport, etc. The regional haze formation in basin areas,
along with intensive emission of precursors, high relative humidity and poor dispersion conditions,
is still limitedly understood.In this study, a field campaign was conducted to investigate the factors
resulting in haze formation in Sichuan Basin (SCB) during winter in 2021. The fine aerosol chemical
composition was characterised by using a time-of-flight aerosol chemical speciation monitor (ToF-
ACSM) with the aim of inorganic and organic aerosol characterisation and source apportionment.
The average concentration of non-refractory fine particles (NR-$PM_{2.5}$) was $98.5 \pm 38.7$ μg/m³, and
organics aerosols (OA), nitrate, sulphate, ammonium, and chloride occupied 40.3, 28.8, 10.6, 15.3
and 5.1 % of $PM_{2.5}$. Three factors, including a hydrocarbon-like OA (HOA), a biomass burning OA
(BBOA), and an oxygenated OA (OOA), were identified by applying the positive matrix



factorisation (PMF) analysis, and they constituted 24.2, 24.2 and 51.6 % of OA on average,
respectively. Nitrate formation was promoted by gas-phase and aqueous-phase oxidation, while
sulphate was mainly formed through aqueous-phase. OOA showed strong dependence on Ox,
demonstrating the contribution of photooxidation to OOA formation. OOA concentration increased
as aerosol liquid water content (ALWC) increased within 200 μg/m$^3$ and kept relatively constant
when ALWC > 200 μg/m$^3$, suggesting the insignificant effect of aqueous-phase reactions on OOA
formation. Among the three haze episodes identified during the whole campaign, the driving factors
were different: the first haze episode (H1) was driven by nitrate formation through photochemical
and aqueous-phase reactions, and the second haze episode (H2) was mainly driven by the intense
emission of primary organic aerosols from biomass burning and vehicle exhaust, while the third
haze episode (H3) was mainly driven by reactions involving nitrate formation and biomass burning
emission. HOA and BBOA were scavenged, while OOA, nitrate, and sulphate formation were
enhanced by aqueous-phase reactions during fog periods, which resulted in the increase of O:C from
pre-fog to post-fog periods. This study revealed the factors driving severe haze formation in SCB,
and implied the benefit of controlling nitrate as well as intense biomass burning and vehicle exhaust
emission to the mitigation of heavy aerosol pollution in this region.



## 1. Introduction

Although a series of emission reduction strategies had been implemented to mitigate severe haze pollution over a decade in China, the fine particle (particulate matter with aerodynamic diameter less than 2.5 μm, PM$_{2.5}$) pollution events still occurred, especially during autumn and winter (Ding et al., 2019; Zhao et al., 2020; Yan et al., 2020). Haze formation was affected by the intense emission of primary particles, rapid formation of secondary aerosols, stagnant meteorological conditions, and topography. The interplay among these factors usually makes haze formation complex (Bao et al., 2019; Guo et al., 2014; Zheng et al., 2015), which resulted in difficulties in making air pollution mitigation strategies.

Many studies showed that the rapid increase of secondary aerosols (including secondary inorganic/organic aerosols, i.e., SIA and SOA) played an important role in haze formation (Huang et al., 2014; Wu et al., 2022). Nitrate, sulphate, and ammonium, mainly formed through photochemical oxidation and aqueous-phase reactions in the atmosphere, were the major component of SIA. Previous studies demonstrated that the substantial formation of nitrate and sulphate exacerbated severe haze development (Wang et al., 2020; Liu et al., 2020). For example, (Zheng et al., 2016) investigated the factors driving haze formation in Beijing in 2013, and the results showed that the enhanced production of sulphate and nitrate led to their increased contribution to PM$_{2.5}$ as the pollution level increased, while the contribution of organic matter (OM) decreased. The emission of SO$_2$ had been reduced dramatically over the past ten years in China; however, NOx did not show a significant reduction. Thus, the haze formation was found to be mainly driven by the reactions generating nitrate in recent years (Fu et al., 2020; Li et al., 2018; Zhou et al., 2021; Zhai et al., 2021).

Compared to SIA, the formation process of SOA was more complicated (Chen et al., 2017). SOA could be formed through the gas-phase photooxidation of volatile organic compounds (VOCs), which was affected by temperature, relative humidity (RH), and total organic aerosol mass loadings (Clark et al., 2016; Hinks et al., 2018; Donahue et al., 2006). SOA could also be formed through the oxidation of water-soluble VOCs or organic products of gas-phase photochemistry, which was observed in the field and laboratory studies (Liu et al., 2018; Chen et al., 2015). Besides, the aging of primary organic aerosols (POA) by oxidants in the atmosphere also contributed to SOA (Wang et al., 2021; Gilardoni et al., 2016). For instance, the organic aerosols emitted from biomass burning



were oxidised through the photochemical oxidation process driven by OH radicals which might take
place in both gas and aqueous phases (Paglione et al., 2020). The variations of regional and seasonal
emissions might also affect the formation of SOA (Dai et al., 2019; Sun et al., 2016). Thus, it is of
vital importance to consider various factors when investigating SOA formation.

Similar to the city clusters such as Beijing, Shanghai and Guangzhou in North China Plain

(NCP), Yangtze River Delta (YRD) and Pearl River Delta (PRD), the Chengdu-Chongqing city
cluster, located in the Sichuan Basin (SCB) in Southwest China, was also suffering severe haze
pollution (Tao et al., 2017; Tan et al., 2019). Many efforts had been made to investigate the temporal
variation, chemical composition and formation mechanism of $PM_{2.5}$ during the evolution of haze
episodes in NCP, YRD and PRD (Peng et al., 2021; Sun et al., 2016; Zhang et al., 2015; Yan et al.,
2020), whereas, only a few studies focused on the factors resulting in haze formation in SCB (Zhang
et al., 2019; Song et al., 2019). These studies mainly focused on the formation of inorganic species
in $PM_{2.5}$, and the results showed that the rapid formation of SIA under high RH conditions promoted
the increase of $PM_{2.5}$. However, further studies on the characteristics of OA, including the temporal
evolution, sources and formation pathways, were still lacking.

The area of SCB is ~260,000 $km^2$, and its population is ~110 million, making it an important

developing region in Southwest China. The basin is a subtropical expanse of low hills and plains
and is completely encircled by high mountains and plateaus, which is unfavourable for either
horizontal transport or vertical diffusion. The atmosphere in SCB was characterised by persistently
high relative humidity and low wind speed all year round. The haze evolution in SCB might be
different from those in other regions due to its unique topography, meteorological conditions and
emission sources, which remained unclear. Therefore, comprehensive studies are needed to reveal
the key factors contributing to haze formation in the basin (Wang et al., 2018).

The Time-of-Flight Aerosol Chemical Speciation Monitor (ToF-ACSM) is a robust and highly

sensitive instrument that provides real-time characterisation and composition of non-refractory
$PM_1/PM_{2.5}$ (NR-$PM_1$/$PM_{2.5}$) depending on the type of the aerodynamic lens used (Liu et al., 2007;
Xu et al., 2017). Compared to the Aerodyne aerosol spectrometer (AMS), the ToF-ACSM is more
compact and cheaper. However, the ToF-ACSM does not figure particle sizing. Compared to the
quadrupole-ACSM (Q-ACSM), the ToF-ACSM has better mass resolution and detection limits
(Fröhlich et al., 2013). Although the ToF-ACSM had not been widely deployed in field observations





as AMS/Q-ACSM did in a number of different sites over China, it had successfully characterised
the variation of $PM_1$/$PM_{2.5}$ and the sources of organic aerosols in the cities within NCP, YRD and
PRD (Ge et al., 2022; Sun et al., 2020; Guo et al., 2020). Despite this, few studies applied a Q/ToF-
ACSM or AMS in SCB. The responses of aerosol chemistry to meteorology and emissions therein
remained poorly understood. To the authors' best knowledge, this is the first time that the ToF-
ACSM has been deployed in the field observations within SCB.

The present study aims to uncover the factors driving severe haze formation during winter in

SBC. The characteristics of NR-$PM_{2.5}$, including mass concentration, chemical composition,
temporal and diurnal variation, were described in detail. The OA sources resolved by the positive
matrix factorisation (PMF) analysis and elemental composition were investigated to elucidate the
possible aging process of OA. The formation mechanism for SIA and SOA was also studied based
on the dependencies of nitrate, sulphate and OOA with odd oxygen ($Ox = O_3 + NO_2$) and aerosol
liquid water content. In addition, the evolution process of chemical composition and elemental
composition were summarised during different haze and fog episodes to investigate the main factors
exacerbating haze pollution. The data and results would fill the knowledge gap on the factors
affecting haze formation in SCB and provide a scientific basis for future air pollutant mitigation
strategies in this unique basin.
**2. Material and methods**
**2.1 Sampling site**

The field study was carried out from 18 December 2021 to 22 January 2022 at a site (30 °55'59"

N, 104 °12'25"E) in Deyang, and the site was near the northern border of Chengdu, the capital of
Sichuan province, China (as shown in Fig. 1). The site was located in a typical suburban region
surrounded by several food, aluminium alloy, and building materials factories. There was a main
road ~200 m south of the site. The north and west of the site were croplands and villages. Thus, the
site was affected by traffic emissions, biomass burning and industrial pollutants. The study at this
site would help to understand the characteristics of regional haze pollution and the influence of
regional transport between urban and suburban areas on haze formation



## 2.2 Instrumentation


During the campaign, the mass loadings of non-refractory organics, nitrate, sulphate,
ammonium and chloride in $PM_{2.5}$ were obtained online by a ToF-ACSM. The gaseous species,
including NO, $NO_2$, $O_3$, $SO_2$, CO and $CO_2$, were continuously measured by the Thermo gas
analysers (model 43i, 49i, 42i, 48i and 410i). The meteorological parameters, including temperature
(T), relative humidity (RH), solar radiation (SR), wind speed (WS) and wind direction (WD), were
obtained by an automatic weather station (Luff WS501-UMB).
For the NR-$PM_{2.5}$ measurement, the ambient air was pumped into the sampling line via a
vacuum pump, and the flow rate was maintained at 3 L/min with a flow meter. Before being sampled
by the ToF-ACSM, the ambient air would go through a $PM_{2.5}$ cyclone (URG-2000-30ED, USA) to
remove coarse particles, then was dried by a Nafion drier. The sampling line was assembled using
3/8' stainless steel tubes coated with the sponge to prevent water condensation.
The working principle of ToF-ACSM had been described in detail in previous studies (Ng et
al., 2011; Fröhlich et al., 2013). Briefly, a 100 μm critical orifice and an aerodynamic lens were
settled in the front inlet system to focus the ambient particles into a concentrated and narrow beam
with a flow rate of ~84 cc/min. It should be mentioned that a $PM_{2.5}$ lens was used during the whole
campaign, which made the $PM_{2.5}$ measurement available (Xu et al., 2017).
The particle beam was transmitted through a vacuum chamber in which the gas-phase species
were separated from the particle beam. At the end of the vacuum chamber, the particles were
thermally vaporised at ~600 ℃ by impacting on a resistively heated porous tungsten surface. There
the non-refractory constituents in the particles flash vaporise and are subsequently ionised by a 70
eV electron impact. Finally, the ions were extracted by a set of ion optics and detected by the time-
of-flight mass spectrometer.
The ToF-ACSM was operated with a time resolution of 10 mins and scanned from m/z 10 to
219. The ionisation efficiency (IE) calibration was performed before and after the campaign
according to the proposal of a previous study (Fröhlich et al., 2013). Briefly, the size-selected (350
nm) $NH_4NO_3$ particles, which were generated by an aerosol generator, were simultaneously sampled
by the ToF-ACSM and a condensation particle counter (CPC 3775, TSI). Then, the IE can be
determined by comparing the response of ToF-ACSM to the mass of ammonium nitrate. The relative





ionisation efficiencies (RIEs) of sulphate and ammonium were also determined by sampling
$(NH_4)_2SO_4$ particles.
**2.3 Data process**
**2.3.1 ToF-ACSM data analysis**
The ToF-ACSM data analysis software (Tofware v2.5.13) written in Igor (Wavemetrics, Lake
Oswego, OR, USA) was used to analyse the mass concentration and chemical composition. The IE
value was 239 ions/pg, and the RIEs for sulphate and ammonium were 1.05 and 3.6, respectively.
For organics, nitrate and chloride, commonly used RIE values, i.e. 1.4, 1.1 and 1.3, were applied.
The algorithm proposed by (Middlebrook et al., 2012) to determine the collection efficiency (CE)
of ToF-ACSM was applied to quantify the aerosol species, as the acidity, chemical composition,
and phase state changed the particle bounce effects at the vaporiser (Matthew et al., 2008).
**2.3.2 PMF analysis**
The mass spectrum data matrix of OA was analysed by the PMF Evaluation Tool (PET, v2.08D)
(Paatero and Tapper, 1994; Ulbrich et al., 2009) in order to resolve distinct OA factors that might
be representative of specific sources. The PMF-ACSM data processing was followed by the
procedures proposed previously (Ulbrich et al., 2009; Zhang et al., 2011). Due to the weak signal
intensities, the ions with m/z over 120 were not included in the analysis. Any ions with signal-to-
noise (S/N) < 0.2 were removed from the analysis, and the ions whose S/N was 0.2–2 were down-
weighted by increasing their errors calculations by a factor of 2 (Sun et al., 2011). Different PMF
solutions were resolved by varying the factor number from 2 to 7. The solutions were evaluated by
comparing the mass spectral profiles of the output secondary aerosol factors as a function of the
rotational parameter (fpeak). Finally, a three-factor solution with fpeak = 0 was selected as our best
solution. The comparison of the results for three to five-factor solutions was described in detail in
Fig. S3 and Table. S2.



### 2.3.3 Identification of haze episodes


The haze episodes were defined by the daily $PM_{2.5}$ mass concentration exceeding Grade II
National Ambient Air Quality Standard (AAQS) of 75 μg/m$^3$ for two successive days (Wang et al.,
2014). In the present study, we made some slight modifications to this definition following the
procedure of Zheng et al. (2016). Briefly, the periods during which the 24 h-moving average
concentration of $PM_{2.5}$ exceeds 75 μg/m$^3$ for two successive days are regarded as episode candidates.
If the hourly $PM_{2.5}$ concentration during the first hour of an episode candidate is 50-75 μg/m$^3$, then
the episode has a shape of 'slow start'; if it is 0-50 μg/m$^3$, then the episode has the shape of the 'rapid
start'. Similarly, the shape at the end of an episode candidate can also be identified. Haze episodes
having a 'slow start' typically arise from the gradual accumulation of pollutants emitted both locally
and regionally under unfavourable meteorological conditions, while haze episodes having a 'rapid
start' are most likely related to regional transport (Zheng et al., 2015). Haze episodes having a 'slow
end' usually resulted from the gradual scavenge of pollutants.
According to the definition mentioned above, three haze episodes (denoted as H1, H2 and H3,
respectively, in Fig. 2), all with the types of 'slow start' and 'slow end', were identified over the whole
campaign. In addition, a fog event occurred during each haze episode (denoted as F1, F2 and F3,
respectively). Lacking the information of visibility and aerosol size distribution up to several tens
of micrometres, we were not able to precisely diagnose the accumulation and dissipation stages of
a fog event. Instead, we selected the hours with RH near 100 % as the duration of a fog event, which
was the same as the condition described in previous studies (Izhar et al., 2020; Guo et al., 2015).

### 2.4 Air mass trajectories analysis


2-day back trajectories arriving at the receptor site were calculated every hour over the whole
campaign using the National Oceanic and Atmospheric Administration (NOAA) HYSPLIT version
4(Draxler and Hess, 1998). Input to the model is in the form of 1 ° latitude-longitude gridded
meteorological parameters from the Global Data Assimilation System meteorological dataset. We
chose an arrival height of 500 m which is above ground level (AGL) for target analysis in the
HYSPLIT model to diminish the effects of surface friction (Polissar et al., 2001); this height value
and greater are regarded as in the open height of the planetary boundary layer in winter and are more



useful for long-range transport. Finally, 839 backward trajectories in total were obtained. Then,
these trajectories were grouped into four clusters, i.e., Cluster1 from the north, Cluster2 from the
southwest, Cluster3 from the northeast, and Cluster4 from the east. The trajectories of each cluster
accounted for 12.6, 6.2, 58.4, and 22.8 % of total air mass trajectories during the whole campaign,
respectively.
**3. Results and discuss**
**3.1 Meteorological condition and chemical composition of NR-PM$_{2.5}$**
**3.1.1 Overview of meteorology and PM$_{2.5}$ chemical composition**

The temporal variation of meteorological parameters, concentrations of gaseous pollutants, and

chemical compositions of PM$_{2.5}$ over the whole campaign are illustrated in Fig. 2. The missing data
were due to the acquisition software malfunction of the instrument. During the campaign, the
temperature ranged from -1.9 to 16.3 ℃ with a mean value of 7.3 ±2.8 ℃, and the RH ranged from
35 to 100 % with an average of 81 ±12.4%. The wind from the southwest prevailed with an average
speed of 0.7 ± 0.5 m/s during the entire campaign. This indicated that the atmosphere was in a
stagnant state with relatively low temperature and high RH.

The mass concentration of NR-PM$_{2.5}$ during the campaign ranged from 23 to 230 μg/m$^3$, with

an average of 98.5 ± 38.7 μg/m$^3$. This was comparable to the average PM$_{2.5}$ concentrations during
wintertime in other cities in the SCB (Table S1). The OA concentration varied from several to 103
μg/m$^3$, with an average of 39.2 ± 3.9 μg/m$^3$, constituting the majority of PM$_{2.5}$. OA contributed ~20-
69 % to PM$_{2.5}$ with an average of 40.3 ±7.6 %. The average concentrations of nitrate, sulphate,
ammonium and chloride were 29 ± 14 (ranging from ~4 to 80 μg/m$^3$), 10 ±4.2 (ranging from ~2 to
28 μg/m$^3$), 15.1 ± 6.4 (ranging from ~3 to 38 μg/m$^3$) and 5.2 ±4.1 μg/m$^3$ (ranging from ~1 to 50
μg/m$^3$), taking up 28.8 ±5.5, 10.6 ±2.8, 15.3 ±2.2 and 5.1 ±3.1 % of PM$_{2.5}$, respectively. The
dominance of organic species was similar to previous observations in urban Chengdu during the
winter of 2014 and 2015 (Kong et al., 2020; Wang et al., 2018). However, the nitrate concentration
was higher than that of sulphate, which was contrary to that reported previously. A recent
observation in urban Chengdu also found that higher fraction of nitrate in PM$_{2.5}$ compared to





sulphate, probably due to the remarkable decrease in $SO_2$ emission in the past ten years (Huang et
al., 2021; Fu et al., 2017).
**3.1.2 Diurnal variation**

The diurnal cycles of $SO_2$, $NO_2$, $O_3$, and $PM_{2.5}$ compositions over the entire campaign were

depicted in Fig. 3. The concentration of $NO_2$ showed a bimodal distribution with peaks at 12:00 and
19:00, most likely due to the emission of vehicles. $SO_2$ concentration had a peak at 12:00, while the
peak concentration of $O_3$ was at 16:00.

Corresponding to the daily temporal variation of $NO_2$, nitrate, and ammonium showed similar

diurnal cycles with two peak concentrations, which indicated the simultaneous formation of these
two species. One of the peaks with a higher concentration showed at 13:00, and the other with a
lower concentration was at 20:00. The formation of nitrate and ammonium could be attributed to
photochemical processing during daytime and heterogeneous reactions during nighttime. Sulphate
showed a peak at 12:00, corresponding to the hour of daily maximum $SO_2$ concentration. The solar
radiation was also near the peak value at this time, suggesting the contribution of intense
photochemistry to sulphate formation (Weber et al., 2016).

Organics and chloride appeared to have similar diurnal cycles. The concentrations of these two

species increased gradually from 6:00 to 10:00 and then decreased till 16:00, which might be
affected by the change of planet boundary (PBL) height. After that, organics increased dramatically,
while chloride increased gradually and reached the second peak at 19:00. The time of these two
peaks were in accordance with rush hours, indicating the possible contribution of traffic emissions
to organics. Chloride was generally regarded as one of the tracers for biomass burning (Chantara et
al., 2019; Vicente et al., 2013). The peaks of chloride in the morning and evening were probably
due to the emission of biomass burning, which would also contribute to the increase of organics.
**3.2 Characteristics of inorganic aerosol**

The correlation between the molar equivalent concentrations of measured ammonium and the

sum of molar equivalent concentrations of nitrate, sulphate and chloride was illustrated in Fig. 4.
The slope of the regression line for ammonium against the sum of nitrate, sulphate and chloride with





a value of 1.01 indicated that the anions in $PM_{2.5}$ were well neutralised by cation (ammonium). This
result illustrated that nitrate, sulphate, and chloride were mainly in the form of $NH_4NO_3$, $(NH_4)_2SO_4$
and $NH_4Cl$, which were commonly considered secondarily formed (Ianniello et al., 2011; Ge et al.,

2017).

As $SO_4^{2-}$ competed with $NO_3^-$ for $NH_4^+$ during their formation, the relationship between nitrate-
to-sulphate molar ratio ($[NO_3^-]/[SO_4^{2-}]$) and ammonium-to-sulphate molar ratio ($[NH_4^+]/[SO_4^{2-}]$)
was indicative of the pathway of nitrate formation (He et al., 2012). If $[NO_3^-]/[SO_4^{2-}]$ linearly
correlated with $[NO_3^-]/[SO_4^{2-}]$ under ammonium-rich conditions ($[NO_3^-]/[SO_4^{2-}] \geq 1.5$), the
homogeneous formation of nitrate was expected:
$$HNO_3 \text{ (g)} + NH_3 \text{ } (g) \leftrightarrow NH_4NO_3 \text{ } (s, aq) \quad (1)$$
While for ammonium-poor conditions ($[NO_3^-]/[SO_4^{2-}] < 1.5$), the high concentration of nitrate
was attributed to its formation through hydrolysis of $N_2O_5$ on the pre-existing aerosols (Pathak et
al., 2009):
$$N_2O_5 \text{ (aq)} + H_2O \text{ } (aq) \leftrightarrow 2NO_3^- \text{ } (aq) + 2H^+ \text{ } (aq) \quad (2)$$
To better elucidate the factors affecting nitrate formation, we divided the observation period
into daytime (6:00 - 18:00, local time) and nighttime (18:00 - 6:00 next day, local time) hours. Fig.
S1 showed that the $[NO_3^-]/[SO_4^{2-}]$ during both daytime and nighttime were larger than 1.5,
indicating ammonium-rich conditions. $[NO_3^-]/[SO_4^{2-}]$ was significantly correlated with
$[NH_4^+]/[SO_4^{2-}]$ during daytime with the regression function:
$$\frac{NO_3^-}{SO_4^{2-}} = 0.69 \times \frac{NH_4^+}{SO_4^{2-}} - 1.08 \quad (3)$$
The intercept of the regression line on the $[NH_4^+]/[SO_4^{2-}]$ axis was 1.56, which was close to
1.5 characterised by (Pathak et al., 2009), suggesting the nitrate formation was mainly driven by
$HNO_3$ production through the reaction of $NO_2 + OH + M \rightarrow HONO_2 + M$. The formation of $HNO_3$
allowed the reaction between $HNO_3$ (g) and excess $NH_3$ (g) to happen, and thus generating
ammonium nitrate (Sun et al., 2011). Indeed, the nitrate concentration and nitrogen oxidation ratio
($NOR = n(NO_3^-)/[n(NO_2) + n(NO_3^-)]$) increased as the Ox concentration increased (as shown in Fig.
5), and exhibited a strong $O_3/Ox$ ratio dependency, which further demonstrated the homogeneous
formation of nitrate during daytime.
During nighttime, the $O_3$ concentration was low and ambient RH was relatively high, which





favoured the aqueous-phase reactions to occur. Higher nitrate concentration was observed with
increasing ALWC (as illustrated in Fig. S2(c)), and so was NOR. This phenomenon further
demonstrated that nitrate was mainly formed through aqueous-phase reactions during nighttime.
The regression function between [NO$_3^-$]/[SO$_4^{2-}$] and [NH$_4^+$]/[SO$_4^{2-}$] was expressed as:
$$\frac{NO_3^-}{SO_4^{2-}} = 0.69 \times \frac{NH_4^+}{SO_4^{2-}} - 1.24 \quad (4)$$

The intercept of the regression line on the [NH$_4^+$]/[SO$_4^{2-}$] axis was 1.80, larger than 1.5. The
regression function suggested that nitrate formation was mainly attributed to homogeneous reactions,
which was not in accordance with the domination of aqueous-phase reactions for nitrate formation
during nighttime as discussed above. It seemed that the intercept of 1.5 for the regression line on
the [NH$_4^+$]/[SO$_4^{2-}$] axis might not be an appropriate proxy to define the formation process of
nocturnal nitrate in the present study, because the emission of NOx and SO$_2$ had been reduced while
NH$_3$ increased in the past almost ten years, which resulted in the ammonium-rich condition in the
atmosphere (Fu et al., 2017; Liu et al., 2018). The abundant NH$_3$ in the atmosphere could
accommodate plenty of basic species during the heterogeneous formation of nitrate, which was
usually considered to occur under ammonium-lean conditions though (Pathak et al., 2009). We
deduced that HNO$_3$ was firstly heterogeneously formed through the hydrolysis of N$_2$O$_5$, then excess
NH$_3$ was uptake by wet particles and neutralised HNO$_3$ forming ammonium nitrate (Man et al.,
2015; Wen et al., 2018). The results were consistent with the study of (Tian et al., 2019), which
demonstrated that heterogeneous hydrolysis of N$_2$O$_5$ dominated nitrate formation during nighttime,
while photochemical reactions played an important role in nitrate formation during daytime.
Fig. S2(b) showed that the average sulphate concentration increased when Ox > 60 μg/m$^3$,
which corresponded to daytime hours, suggesting the contribution of the photochemical process to
sulphate formation. However, the sulphur oxidation ratio (SOR = n(SO$_4^{2-}$)/[n(SO$_2$) + n(SO$_4^{2-}$)])
decreased with increasing Ox, suggesting the photooxidation was not efficient for converting SO$_2$
to sulphate. By contrast, SOR showed an increasing trend as ALWC increased, demonstrating the
efficient conversion of SO$_2$ to sulphate through aqueous-phase reactions. A previous study showed
that the aqueous oxidation of SO$_2$ by NO$_2$ is key to efficient sulphate formation on fine aerosols
with high relative humidity and NH$_3$ neutralisation (Wang et al., 2016). As mentioned above, the
atmospheric aerosols were well neutralised, and RH was high, which favoured the following



reaction to occur:
$$SO_2\,(g) + 2NO_2\,(g) + 2H_2O\,(aq) \rightarrow 2H^+(aq) + SO_4^{2-}(aq) + 2HONO(g) \quad (5)$$
**3.3 Characteristics of organic aerosol**
**3.3.1 Source appointment of OA**
PMF analysis was performed to explore the OA sources measured during the whole campaign.
A three-factor solution was chosen as the best PMF analysis results based on the mass spectra profile,
variation of Q/Qexp, diurnal variation and correlation with external tracers. The resolved factors
included a hydrocarbon-like OA (HOA), a biomass burning OA (BBOA) and an oxygenated OA
(OOA). The mass spectra profiles of these three factors were shown in Fig. 6, and the temporal
profile of each factor and its external tracer were also shown.
The mass spectrum of HOA was dominated by the ions of m/z 29, 41, 55, 57, 69 and 71. HOA
was well correlated with BC (r = 0.77), which was largely emitted by vehicles. The diurnal cycles
of HOA showed two peaks during typical rush hours, demonstrating the contribution of traffic
emissions to HOA (Lanz et al., 2007; Zhang et al., 2005). The average concentration of HOA was
8.9 ± 6.5 μg/m³ and constituted 24.2 ± 10.4 % of OA over the whole campaign. The increased
fraction of HOA as a function of total OA (Fig. 7) demonstrated the contribution of motor vehicle
emissions to haze formation.
BBOA was characterised by the pronounced peaks at m/z 60 (mainly $C_2H_4O_2^+$) and 73
($C_3H_5O_2^+$), which were generally regarded as biomass burning markers from levoglucosan
compounds (Mohr et al., 2012; Weimer et al., 2008; Alfarra et al., 2007). BBOA was well correlated
with m/z 60 and m/z 73 (r = 0.76 and 0.94, respectively) and accounted for 88 % of m/z 60 and 70 %
for m/z 73, which were higher than those in other sources. A good correlation was also found
between BBOA and chloride (r = 0.64), which was also suggested to be one of the tracers of biomass
burning. The diurnal variation of BBOA showed a similar trend of chloride, with two peaks during
8:00-10:00 and at 19:00, which was due to the residential cooking and heating using biomass. The
concentration of BBOA ranged from ~1 to 34 μg/m³, with an average of 8.9 ± 5.4 μg/m³. BBOA
took up 24.2 ±8.6 % with a maximum of 46 % of OA, and its fraction also increased with increasing



total OA concentration, indicating the contributions of biomass burning activities during haze
episodes.
OOA was featured by the dominant signal intensities at m/z 28 (mainly $CO^+$) and m/z 44 ($CO_2^+$).
OOA accounted for 69 % of m/z 44, which was higher than those in other sources. The time series
of OOA correlated well with those of nitrate and sulphate (r = 0.81 and 0.72, respectively),
suggesting the commonly regional and aged properties of OOA. The concentration of OOA
accumulated gradually from 8:00 to 13:00, then decreased till night time. The diurnal cycle of OOA
was similar to solar radiation, suggesting OOA formation was associated with photochemical
reactions. The average OOA concentration showed increasing trends as Ox concentration increased
during both daytime and nighttime (Fig. S4), indicating the probable formation pathways of OOA
from its precursors (Kuang et al., 2020; Zhan et al., 2021). Note that the accumulation of $NO_2$ would
lead to the decrease of $O_3$/Ox ratio at night; thus, Ox might not be an appropriate indicator of
photochemical oxidation (Xu et al., 2017; Herndon et al., 2008). The OOA formation might be
attributed to other processes during nighttime. For example, previous studies showed that high NOx
concentration facilitated the formation of nitrate radical ($NO_3$), and the $NO_3$ oxidation of biogenic
volatile organic compounds (BVOC) was important for nighttime secondary organic aerosol
formation (Boyd et al., 2017; Rollins et al., 2012). The average OOA concentration did not change
significantly with increasing ALWC during daytime, suggesting the less contribution of aqueous-
state reaction to the formation of OOA. During nighttime, the average OOA concentration showed
an increasing trend when ALWC < 200 $\mu g/m^3$ and kept relatively constant subsequently, suggesting
the aqueous-phase reactions did not significantly affect the formation of OOA.
**3.3.2 Elemental composition of OA**
The $f$44 vs. $f$43 and $f$44 vs. $f$60 during the entire campaign were illustrated in Fig. 8. The triangle
plot of $f$44 vs. $f$43 has been widely used to characterise OA evolution in the atmosphere because
m/z's 44 and 43 are usually from different functional groups, and the ratio changes as a function of
atmospheric aging. The $f$43 ranged from ~0.06 to 0.12 with an average of 0.08 ± 0.009, and the
range of $f$44 was ~0.07-0.24 with an average of 0.15 ± 0.03, suggesting the existence of both fresh
and aged aerosols in the atmosphere. Most of the data were within the triangle space characterised



by a series of field observations and experimental data (Ng et al., 2011). However, those points with
higher $f44$ were outside the upper boundary of the triangle region, suggesting the corresponding
aerosols were more oxidised. It could be observed that the points with higher $f44$ (> 0.16)
corresponded to relatively higher Ox concentration and lower RH, while those with lower $f44$
corresponded to relatively lower Ox concentration and higher RH, suggesting that the formation of
more oxidised OOA was mainly attributed to photochemical reactions, and the formation of less
oxidised OOA was probably attributed to aqueous-state reactions (Zhao et al., 2019; Kim et al.,

2019).

The triangle plot of $f44$ vs. $f60$ was widely used as a metric to access the evolution of
atmospheric BBOA (Cubison et al., 2011). A value of 0.003 for $f60$ was recommended as an
appropriate value to represent the atmospheric background free of biomass burning influence (Aiken
et al., 2009; Docherty et al., 2008). The $f60$ ranged from 0.0028 to 0.055 with an average of 0.008
±0.003 during the campaign. Except for several points, the $f60$ was ubiquitously higher than 0.003
and most points fell in the triangular region, suggesting the contribution of biomass burning to OA.
The $f44$ and $f60$ of BBOA resolved by PMF in the present study were also in the triangular region
and comparable with previous studies (Paglione et al., 2020; Zhao et al., 2019; Kim et al., 2019).
Fig. S5 showed that $f44$ increased while $f60$ decreased with increasing Ox, indicating the likely
oxidation of levoglucosan and/or levoglucosan-like substances. Cubison et al. (2011) suggested that
the increasing $f44$ was not only attributed to the oxidation of levoglucosan-like species, but the
oxidation of bulk OA also played a role because the levoglucosan-like species only contributed a
small fraction of the OA mass (Aiken et al., 2009). Their contribution to the total signal m/z 44
before or after aging was also small. Compared to the effects of Ox, the increasing RH did not seem
to push $f60$ to the left upper region. Higher RH was observed for those data points within the region
of aged BBOA in the $f44$ vs. $f60$ space, as reported previously by Paglione et al. (2020), which
indicated the probable aqueous-phase oxidation of BBOA. Although the aqueous OOA (aq-OOA)
could not be resolved in the present study, we deduced that the aqueous-phase reactions occurred
and contributed to the formation of OOA, for (1) the ambient RH was typically above 65%, which
favoured the presence of wet aerosol particles; (2) the range of O/C estimated from the Improved-
Ambient (IA) method varied from 0.46 to 0.85 when the ambient RH > 80 %, covering the O/C
range of the OOA obtained from the photooxidation of organic precursors in the aqueous phase and





ambient aq-OOA observed in many other cities (Duan et al., 2021; Mandariya et al., 2019; Sun et
al., 2010; Xu et al., 2017; Gilardoni et al., 2016).

The evolution of OA during the whole campaign was characterised by the van Krevelen (VK)

diagram in Fig. 9. The slope obtained from the linear regression of H:C versus O:C plotted in the
VK-$\overline{OS_c}$ space could be used to infer the composition of OA and the chemical process in OA
formation (Docherty et al., 2018). The slope of 0 in VK-$\overline{OS_c}$ plot was related to the replacement of
a hydrogen atom with an OH moiety, while the slopes of -0.5 and -1 indicated the formation of
carboxylic groups with/without fragmentation, and a slope of -2 was equivalent to the replacement
of an aliphatic carbon with a carbonyl group (Heald et al., 2010; Ng et al., 2011). The slope of the
linear fitting line for all the data points was -0.14, suggesting the probable formation of carboxylic
acid moieties and hydroxyl groups. The slopes of the linear fitting lines for each fog episode were similar
and close to zero, which was consistent with the hydroxyl group formation possibly taking place in
aerosol water through dark chemistry (Sun et al., 2010; Yu et al., 2014; Zhao et al., 2014).

## 3.4 Case studies for haze pollution

### 3.4.1 Factors driving the evolution of haze episodes

As mentioned above, three haze episodes were identified over the whole campaign. The

synoptic conditions and aerosol chemical composition for each haze episode were summarised in
Table. S2. The average temperature during H2 was lower than those during H1 and H3, while the
average SR was higher. The mean RH and wind speed were almost the same during each haze
episode. The average concentrations of aerosol chemical composition and their contributions to
PM$_{2.5}$ were different in each haze episode, indicating that the factors causing haze formation might
be different during the campaign.

The average PM$_{2.5}$ concentrations measured by ToF-ACSM during H1, H2, and H3 were 113

±46, 109 ±46, and 104 ± 30 μg/m$^3$, respectively. The average mass fractions of OA, nitrate, sulphate,
ammonium, and chloride were similar during H1 and H3 (as shown in Fig. 10(a)). During H2, the
mass fractions of sulphate and ammonium were slightly higher than those in H1 and H3, while the
mass fractions of OA and chloride were lower. In OA, the fraction of primary organic aerosols (POA
= HOA + BBOA) during H3 was higher than those during H1 and H2. The fraction of BBOA showed



an increasing trend from H1 to H3, demonstrating the contribution of biomass burning to haze
formation. Despite the importance of BBOA to winter haze formation in SCB, the control of biomass
burning did not receive much attention, and more efforts were needed for atmospheric aerosol
mitigation in the future.

Fig. 10(b) showed that the concentrations of OA, nitrate, sulphate, ammonium, and chloride

all increased as the ambient air quality worsened (implying by the increasing $PM_{2.5}$ concentration)
during each haze episode. During H1, the fraction of nitrate in $PM_{2.5}$ increased, while the proportions
of sulphate, OOA, and HOA in $PM_{2.5}$ decreased as the $PM_{2.5}$ concentration increased, indicating the
evolution of this haze episode was mainly driven by the reactions involving nitrate formation. For
H2, the fraction of nitrate and ammonium in $PM_{2.5}$ did not show apparent changes, and the fractions
of sulphate and OOA decreased, while the fractions of BBOA and HOA increased as the air quality
worsened, demonstrating that the emission of primary organic aerosols from biomass burning and
vehicle exhaust were the major factors which drove the haze formation. During H3, the fractions of
nitrate and BBOA in $PM_{2.5}$ increased, while OOA decreased and the rest composition did not change
significantly as the $PM_{2.5}$ concentration increased, showing that the evolution of this haze episode
was mainly driven by the reactions involving nitrate formation and biomass burning.

The average estimated O:C and H:C during H1 and H2 were similar and slightly higher/lower

than that during H3, thus resulting in a higher carbon oxidation state ($\overline{OS_c} \approx $ 2O:C - H:C) during
H1 and H2. The lower average $\overline{OS_c}$ during H3 might be due to the higher proportion of HOA and
BBOA, which did not undergo long-time aging and kept relatively fresh in the atmosphere.

**3.4.2 Regional transport**

Air mass from the north (Cluster1) transported at relatively high heights before arriving at the

observation site (as illustrated in Fig. 11), and corresponded to the lowest average $PM_{2.5}$
concentration (66 ± 30 μg/m³). The air mass of Clsuter2 had the longest transport distance and
highest transport height. Although they took up the least proportion of total air mass, they had the
highest average $PM_{2.5}$ concentration (119 ± 30 μg/m³) during the whole campaign, suggesting that
pollutants accumulated high in the air. Air parcels from the east (Cluster3) with the shortest transport
distance and relatively low transport height had an average $PM_{2.5}$ concentration of 113 ± 34 μg/m³.


This indicated that the pollutants might be brought to the observation site along with the transport
of air mass originating from adjacent areas. The air mass from the northeast of Sichuan province
(Cluser4) had the lowest transport height; however, the corresponding PM$_{2.5}$ concentration was
lower than that of Cluster3.
Compared to cluster1 and cluster4, a higher contribution of nitrate to PM$_{2.5}$ was observed for
Cluster2 and Cluster3, which was mainly related to the intense emission from industry and vehicles.
However, the contribution of BBOA was higher for Cluster3 and Cluster4 compared to other clusters,
suggesting intense biomass burning along their transport paths.
To better understand the potential pollutant sources during the campaign, the potential source
contribution function (PSCF) was applied to analyse the possible regions that might contribute to
haze formation. The spatial distribution of weighted PSCF for different chemical compositions was
illustrated in Fig. 12. The WPSCF values for organics over the southwest and southeast were > 0.7,
indicating these locations were likely source areas of organics. For nitrate and sulphate, the areas
by the southwest and south of the sampling site were potential source regions. The major source
regions for HOA were the areas of south Sichuan province and southwest Deyang. The regions
contributing to BBOA were the areas of east Sichuan province, which was consistent with a higher
proportion of BBOA in PM$_{2.5}$ in the air mass stemming from these locations. For OOA, the WPSCF
values over the southwest and southeast of Deyang and northeast Sichuan province were > 0.7,
suggesting the contribution of these regions.

## 3.5 Evolution of chemical composition during fog periods

The fog periods usually started at night or early morning and dissipated in the afternoon. The
individual meteorological parameter differed among each fog period (as summarised in Table S3).
The average temperature was the highest during F1 (6.2 ± 2.3 ℃), and the lowest was observed
during F2 (1.5 ± 3.2 ℃). The maximum solar radiation during F1 and F2 (470 and 500 W/m$^2$) were
similar and much higher than that during F3 (75 W/m$^2$). The synoptic conditions with low
temperature and calm wind favoured the formation of radiation fogs in each fog period.
The chemical composition of PM$_{2.5}$ was also different during each fog period. The average
concentrations of organics, nitrate, and ammonium were almost the same during F2 and F3 and



significantly lower than those during F1. However, the average chloride concentration during F2
was twice of those during F1 and F3, suggesting the possibility of stronger biomass burning
emissions during F2. OOA constituted the major part of OA during F1, while HOA and BBOA were
more important than OOA during F2 and F3. The domination of secondary species in $PM_{2.5}$ during
F1 was probably due to the aqueous-phase reaction, while the primary emission tended to be
stronger during F2 and F3.
Since the aerosols were dried by a Nafion drier, the aerosols that ToF-ACSM measured were
the interstitial particles in droplets or those excluded from fog droplets. In order to better characterise
the evolution of chemical composition in each fog period, the intervals for 3 h before and after the
fog period, when ambient RH was lower than 95 %, were regarded as pre-fog and post-fog periods,
respectively (Kim et al., 2019). Note that the post-fog was not assigned for F3 because the ToF-
ACSM data were not available with a failure of acquisition software.
The average concentrations of different chemical compositions during the evolution of fog
episodes (pre-fog, during fog, and post-fog) were illustrated in Fig. 13. Compared to the pre-fog
period of F1, all species in the interstitial aerosols decreased during the foggy period, likely due to
the scavenging by fog droplets. Note that the OOA and sulphate decreased less than other species,
probably due to the OOA and sulphate formation through aqueous-phase reactions against the
scavenging effect of fog droplets. Except for OOA and nitrate, all species kept decreasing during
post-fog periods, which might be attributed to the increase of the PBL height. The increase of OOA
and nitrate was probably associated with the enhancement of photochemical reaction after the
dissipation of fog.
Distinguished from F1, all species (except for HOA) increased during the foggy period from
the pre-fog period during F2. Although hydrophilic species (e.g., nitrate, sulphate, and OOA) tended
to be scavenged by fog droplets, it seemed that the formation of OOA, nitrate, sulphate, and
ammonium was significantly faster than wet removal, thus resulting in an increase during the foggy
period. BBOA and HOA were commonly considered as hydrophobic species, and they were
excluded from fog droplets. Previous studies showed that BBOA and HOA concentrations decreased
during the foggy period compared to the pre-fog period (Collett et al., 2008; Kim et al., 2019),
despite their insoluble nature. The increase of BBOA in the present study was attributed to the
intense emission from biomass burning during the foggy period, which overwhelmed the scavenging



effects of fog droplets. During the post-fog period, BBOA and HOA decreased significantly,
possibly due to weaker emission and the efficient removal of fog droplets through nucleation and/or
coagulation. The dynamics of PBL might also play a role because the temperature kept increasing,
and higher PBL was expected during the post-fog period. With stronger solar radiation, OOA, nitrate,
and ammonium continued growing during the post-fog period from the foggy period through
photochemical reactions. However, sulphate slightly decreased, which might be due to the
insufficient formation through photochemical reactions, and decreased as the PBL height increased.
Similar to F2, all the secondary species increased during the foggy period from the pre-fog period
during F3. However, BBOA and HOA were reduced significantly by the scavenging of fog droplets.

The average elemental O:C showed an increasing trend from pre-fog periods to post-fog/foggy

periods, while H:C did not change significantly for different fog events, suggesting the OA became
more oxidised. As shown in Fig. S6, the mass fractions of OOA increased, while the contribution of
BBOA and HOA decreased from pre-fog periods to post-fog/foggy periods for the three fog events.
As a consequence, the O:C increased in line with the increased contribution of OOA. Previous
studies had reported that the aqueous-phase reactions enhanced OOA formation during fog/high RH
periods (Chakraborty et al., 2016; Kuang et al., 2020; Chakraborty et al., 2015). It appeared that the
OOA formation would balance out the scavenging of fog droplets during the foggy period for the
three fog events, despite the hydrophilic OOA being preferentially scavenged by fog droplets. Thus,
the OOA concentration marginally decreased or even increased during foggy periods.
**4. Conclusions**

Haze pollution has long been a severe environmental problem in SCB. The formation process

of haze pollution in SCB might be different from those in NCP, YRD, and PRD due to the unique
topography and meteorological conditions, which are still unclear. Based on the measurement data
of a ToF-ACSM and combined with the PMF and PSCF analysis, the temporal variation, formation
process, and sources of $PM_{2.5}$ were characterised to elucidate further the factors contributing to haze
formation. It was found that the concentrations of OA and nitrate increased dramatically as $PM_{2.5}$
concentration increased, and the stagnant synoptic condition favoured the accumulation of these
pollutants. For different haze events, the driving factors could be classified into three types: one was



the reactions involving nitrate formation; another one was the intense biomass burning and vehicle
exhaust emissions, and the last one was the combination of the reactions involving nitrate formation
and biomass burning. Nitrate formation was primarily affected by photooxidation during daytime,
while the nocturnal nitrate formation was dominated by aqueous-phase reactions. OOA constituted
a major part of OA, and it was mainly generated through photooxidation, while aqueous-phase
reactions did not significantly promote its formation.
Due to the limitation of the present study, the parameters which are indicative of the pathways
of nitrate formation are not characterised. The major precursors contributing to a large amount of
OOA are not clear yet. In addition, how controlling BBOA will affect the atmospheric visibility,
radiative forcing, and climate change in SCB needs further investigation in the future. In spite of
the deficiencies, the results in this study implied that controlling primary emissions (such as biomass
burning and vehicle exhaust) and precursors of secondary aerosols (e.g., $NO_x$, $SO_2$, and VOCs)
during severe haze periods will benefit the improvement of air quality in SCB.




*Data availability.* The data generated and analysed in this study are available on visiting
https://doi.org/10.5281/zenodo.6965551.

*Author contributions.* FY and SX designed this study. XZ and QL contributed to data collection
during the field campaign. JZ, DZ, and CZ performed field experiments. ZL and CP performed the
data analysis. ZB wrote the manuscript. GS, LZ, and YC contributed to the scientific discussion and
paper correction.

*Competing interests.* The authors declare that they have no conflict of interest.

*Acknowledgements.* This work was supported by the National Key Research and Development Program of China
(grant no. 2018YFC0214001), and the National Natural Science Foundation of China (grant no. 42075109).

*Financial support.* This research has been supported by the National Key Research and Development Program of
China (grant no. 2018YFC0214001), and the National Natural Science Foundation of China (grant no. 42075109).





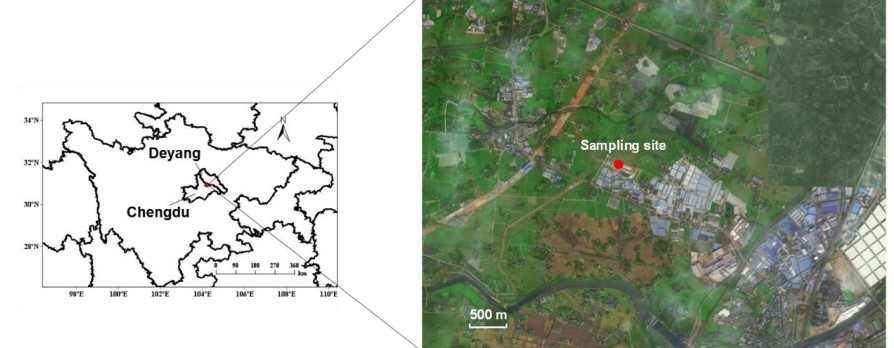


Fig. 1 Location of the observation site (from Baidu Maps, ©2022 Baidu – GS(2021)6026).





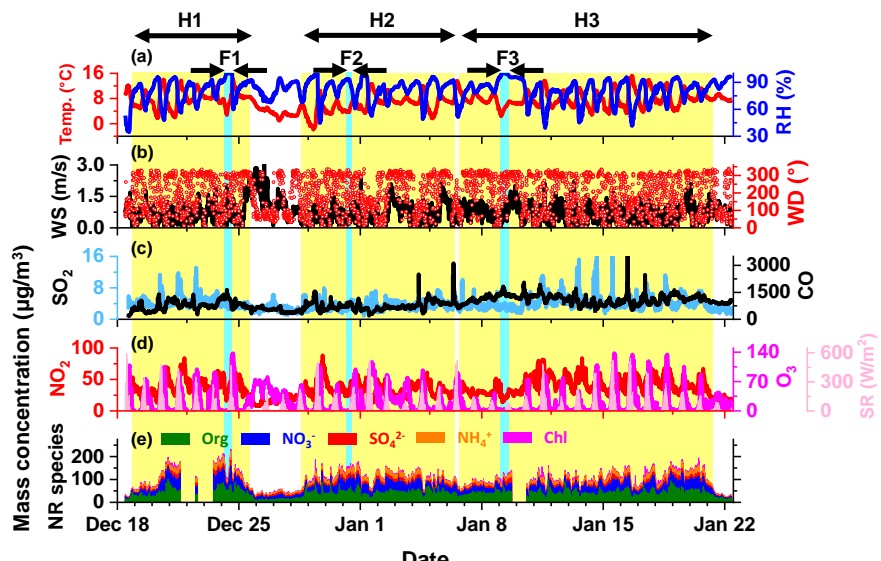


Fig. 2 Time series of (a) relative humidity (RH) and temperature (T); (b) wind direction and wind
speed; (c), (d) CO, $NO_2$, $SO_2$, and $O_3$ mass concentrations and solar radiation; (e) mass concentration
of organics, nitrate, sulphate, ammonium, and chloride in NR-$PM_{2.5}$. The yellow-shaded areas
represent the intervals of H1, H2, and H3, respectively. The light blue-shaded areas represent the
intervals of F1, F2, and F3, respectively.





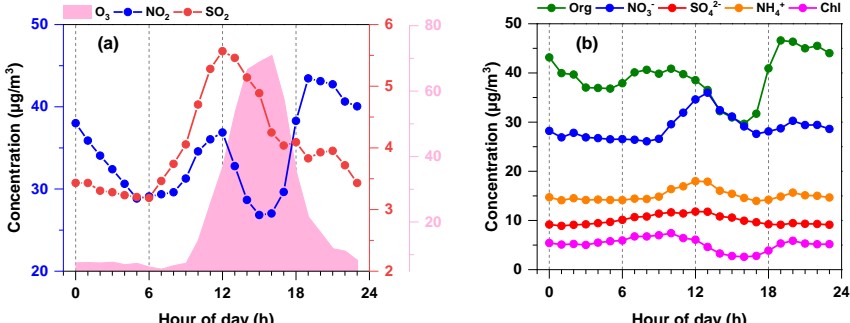


Fig. 3 Diurnal variations of (a) $O_3$, $NO_2$, and $SO_2$, (b) chemical composition in NR-PM$_{2.5}$.








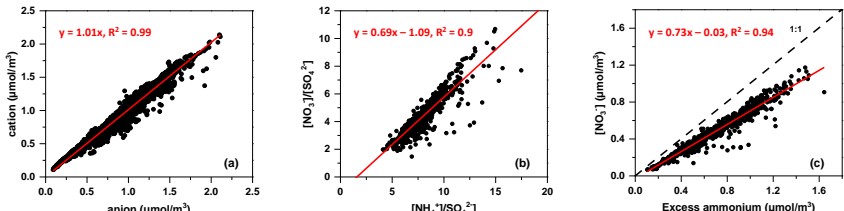


Fig. 4 Scatter plots of (a) molar concentrations of cations vs. anions, (b) molar ratios of nitrate to

sulphate vs. ammonium to sulphate, and (c) molar concentrations of nitrate vs. excess ammonium.






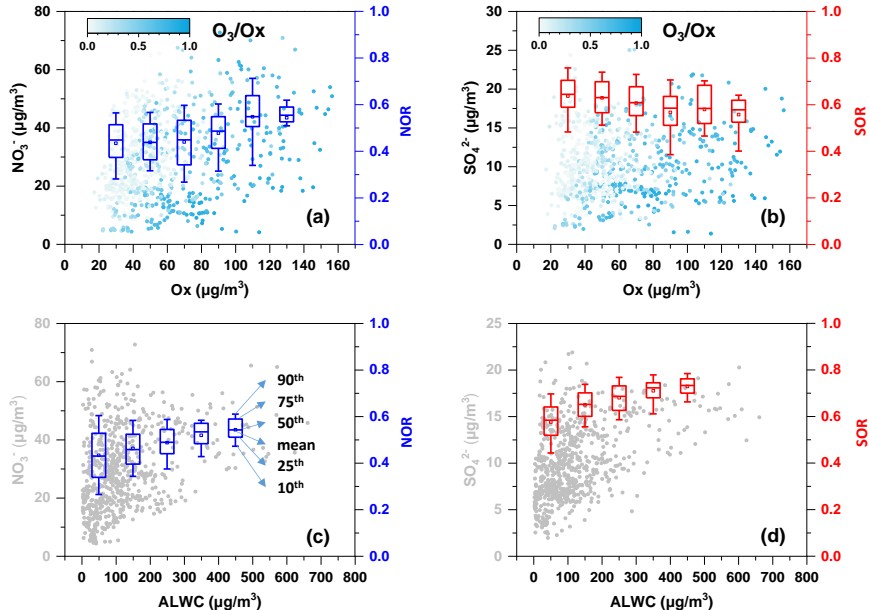


Fig. 5 Variation of (a), (c) nitrate and NOR and (b), (d) sulphate and SOR as Ox/ALWC increases.

The data NOR and SOR were grouped into different bins according to 20 μg/m$^3$ increment of Ox in

(a) and (b), and 100 μg/m$^3$ increment of ALWC in (c) and (d). The colour scale represents O$_3$/Ox

ratios in (a) and (b). The mean (square), 50th (horizontal line inside the box), 25th and 75th

percentiles (lower and upper box), and 10th and 90th percentiles (lower and upper whiskers) of the

box chart are marked in (c). The concentration of ALWC was simulated using the ISORROPIA-II model.








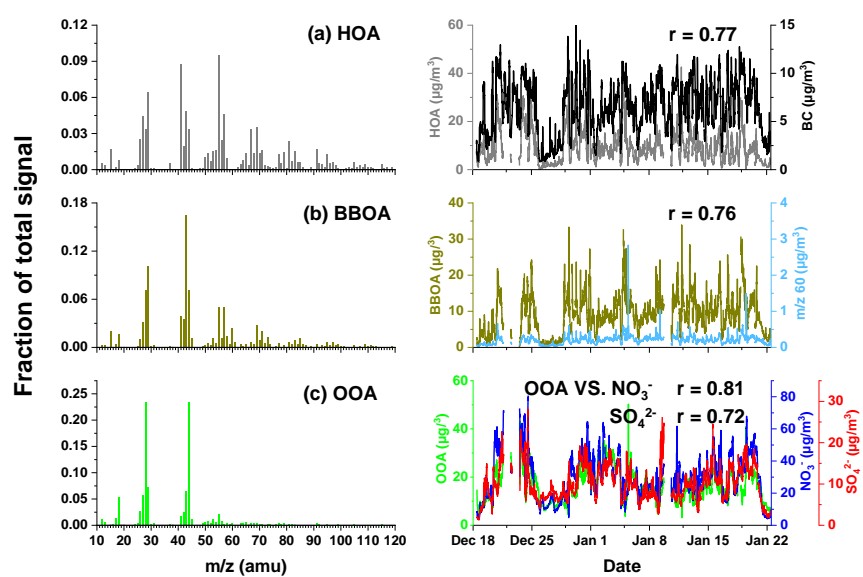


Fig. 6 Mass spectrum of (a) HOA, (b) BBOA, and (c) OOA resolved by PMF. The time series of

each OA source and corresponding tracers are depicted in the right panel.




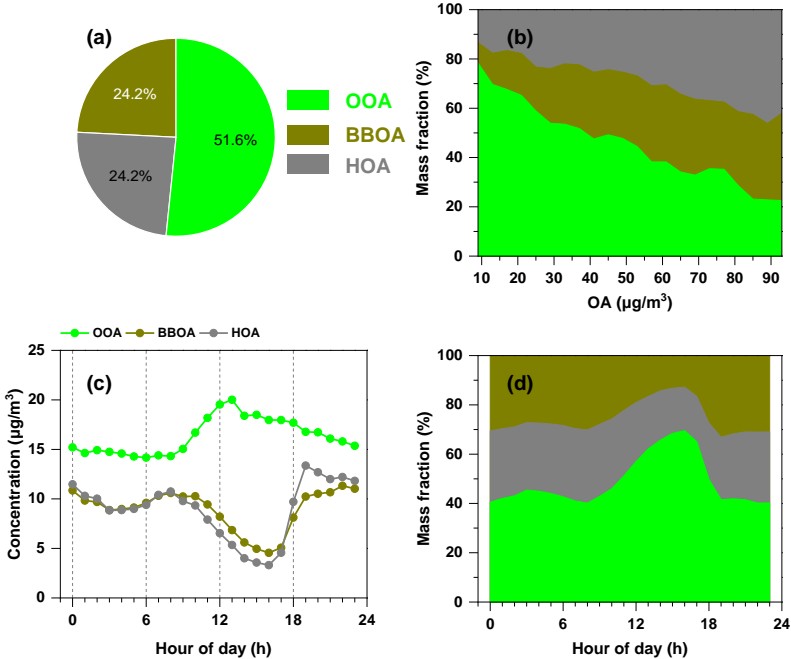


Fig. 7 Average mass fraction of different OOA, BBOA, and HOA (a) in OA and (b) as a function of

OA mass concentration. The diurnal variation of different OA compositions and their mass

contributions are shown in (c) and (d).



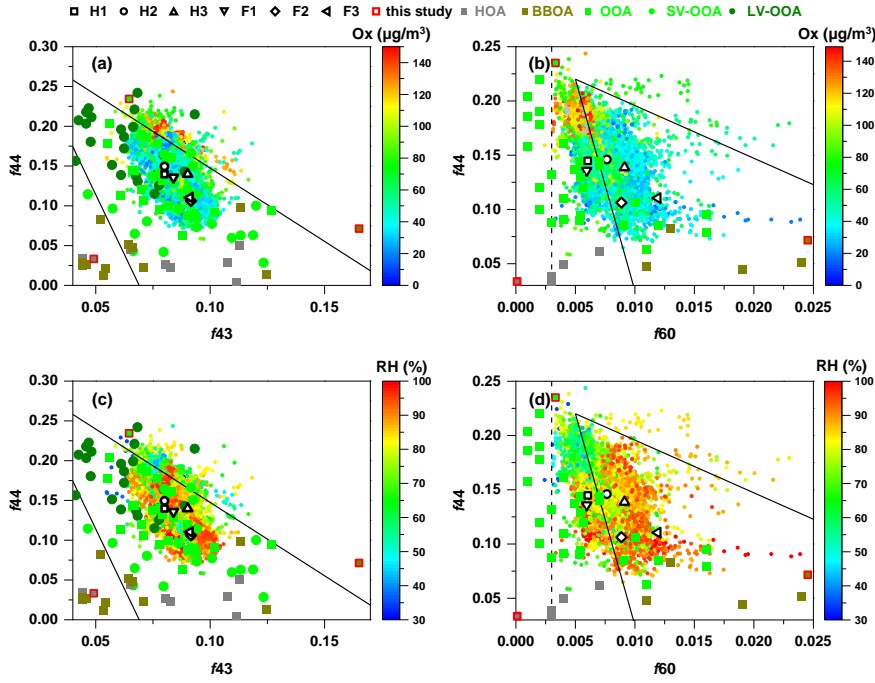

Fig. 8 Triangle plots of (a), (c) $f44$ (ratio of m/z 44 to total signal in the component mass spectrum)

vs. $f43$ (ratio of m/z 43 to total signal in the component mass spectrum), and (b), (d) $f44$ vs. $f60$

(ratio of m/z 60 to total signal in the component mass spectrum) during the whole campaign. The

colour scale in (a) and (b) represents Ox concentration, and that in (c) and (d) represents RH. The

solid lines in (a) and (c) are derived from the results reported by Ng et al. (2010). The dashed line

representing the background value of secondary aged OA and the solid guidelines in (b) and (d) are

derived from Cubison et al. (2011). The $f44$ vs. $f43$ and $f44$ vs. $f60$ for different OA sources reported

in previous studies are also shown (Kim et al., 2019; Ng et al., 2011; Wang et al., 2016; Zhao et al.,

2019; Paglione et al., 2020).






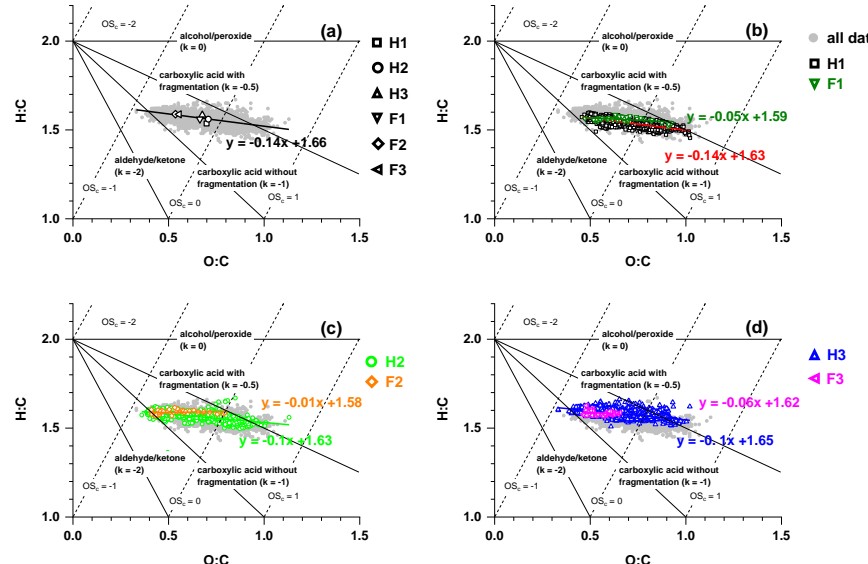


Fig. 9 The Van Krevelen-$\overline{OS_c}$ diagram with data during (a) the whole campaign, (b) H1 and F1, (c)

H2 and F2, and (d) H3 and F3. The triangle lines with different slopes show distinct formation

processes (Heald et al., 2010).




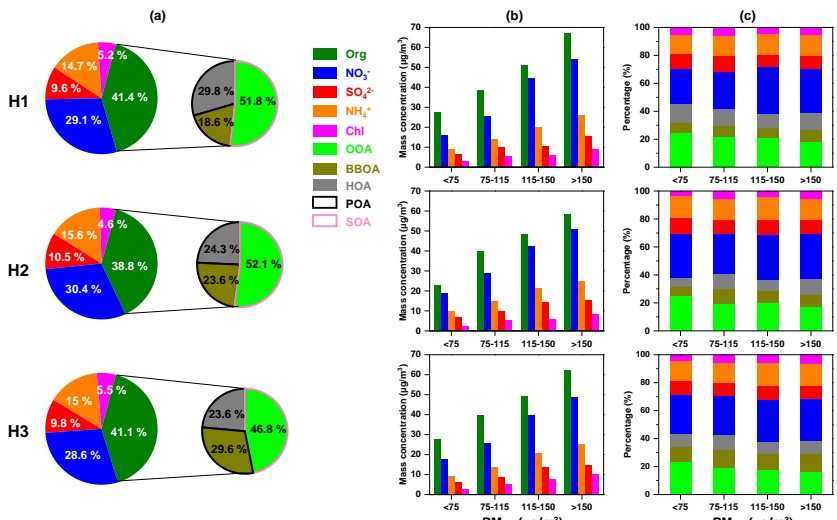


Fig. 10 (a) Average mass fractions of different chemical compositions in PM$_{2.5}$, (b) mass
concentration, and (c) relative contribution of inorganic/organic species as a function of PM$_{2.5}$
concentration during each haze episode. The right panel in (a) depicts the contribution of OOA,
BBOA, and HOA to OA.



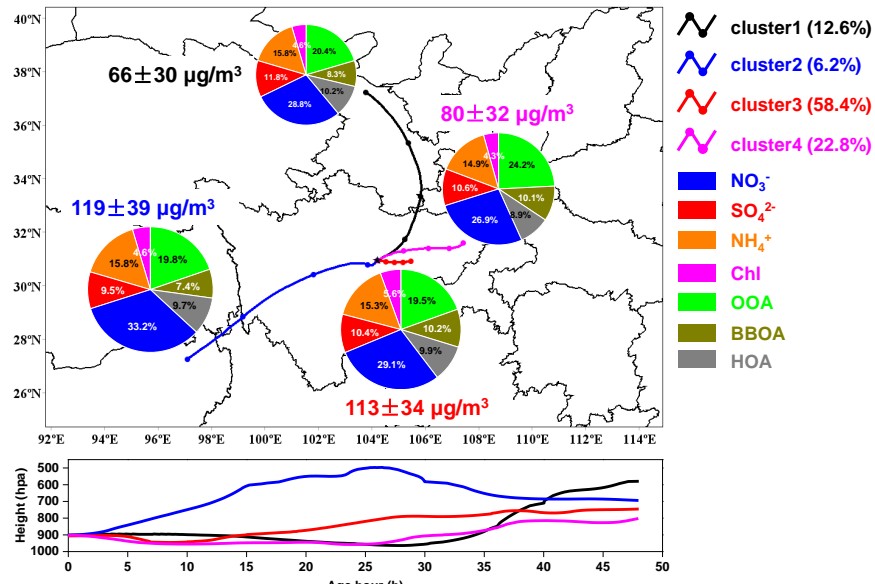


Fig. 11 Simulation results of 48 h backward air mass cluster-mean trajectories during the campaign.
The lines in black, blue, red, and purple represent the mean trajectories of Cluster1 to Cluster4,
respectively. The pie charts show the average mass contribution of different chemical compositions
to PM$_{2.5}$ for each cluster. The lower panel shows the height profile for different air mass clusters along
their transport paths.




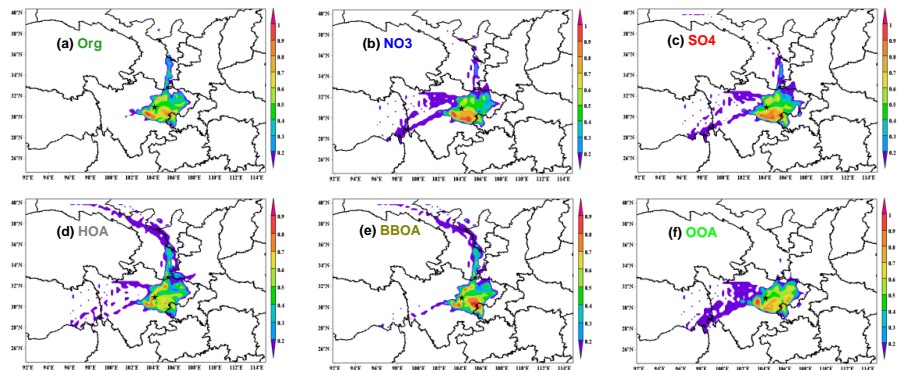


Fig. 12 Simulation results of PSCF for (a) organics, (b) nitrate, (c) sulphate, (d) HOA, (e) BBOA,
and (f) OOA during the whole campaign.





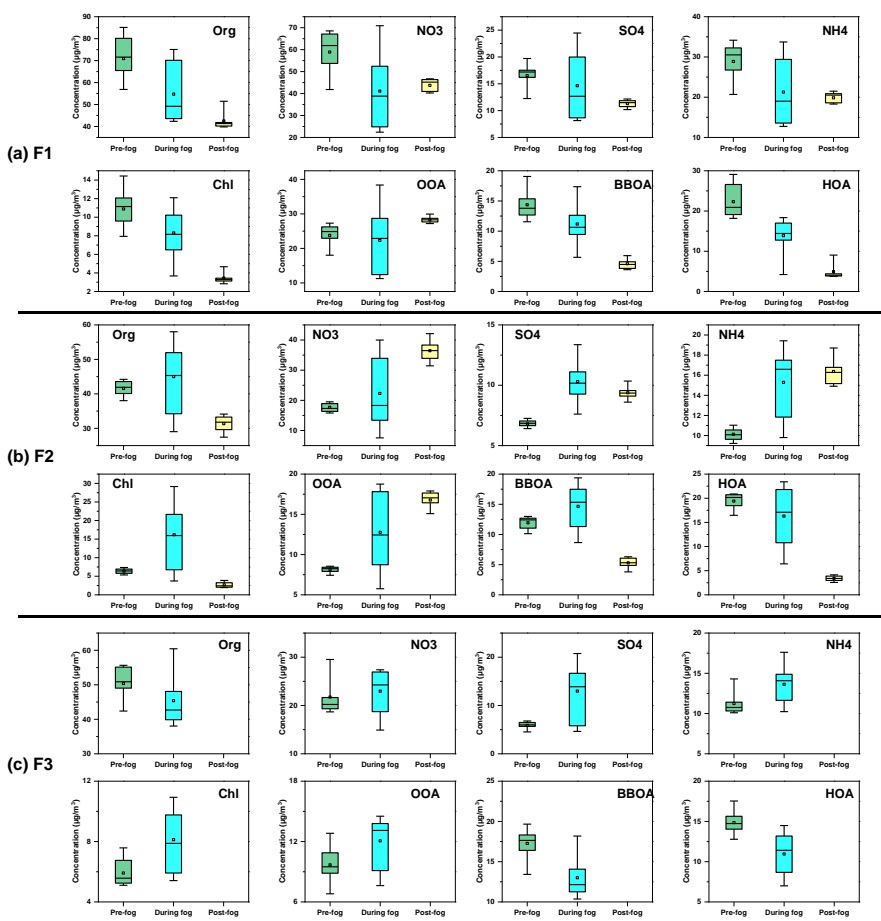


Fig. 13 Variation of organics, nitrate, sulphate, ammonium, chloride, OOA, BBOA, and HOA

concentration during the evolution of (a) F1, (b) F2, and (c) F3.







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
