# Peer review of "Measurement report: Intensive biomass burning emissions and rapid nitrate formation drive severe haze formation in Sichuan basin, China: insights from aerosol mass spectrometry"

_Atmospheric Chemistry and Physics, 2022_

## Author Response (AR1)

Manuscript No. ACP-2022-477

Tittle: Measurement report: Intensive biomass burning emissions and rapid nitrate formation drive severe haze formation in Sichuan basin, China: insights from aerosol mass spectrometry

The authors gratefully thank all the reviewers for their comments and suggestions. We have revised our manuscript according to the two reviewers' suggestions and comments. **All the changes and responses to the reviewers' comments are listed below point-by-point. The changes are highlighted with red in the revised manuscript.** We sincerely hope this manuscript will be acceptable for publication in *Atmospheric Chemistry and Physics*.

Comments from the reviewers:

Reviewer1:

The manuscript by Bao et al. present detailed observations of the chemical composition of PM2.5 in the Sichuan Basin (SCB), and the component responsible for the formation of haze during winter. The PM2.5 composition on site is driven by gas phase and aqueous-phase oxidation for nitrate, aqueous phase formation for sulfate, primary emission from Biomass burning and vehicle emissions and nitrate formation influenced by biomass burning. During fog events, primary organic aerosols were scavenged while secondary aerosol formation was enhanced by aqueous-phase reactions. The method applied and the case studies presented provide valuable knowledge on the species and mechanisms leading to haze and fog events in the SCB, but some restructuring and improvement in the discussion need to be addressed before publication.

General comments:

Page 11 lines 296-316: The discussion about night time nitrate formation is a bit confusing or not well constructed. During daytime, you attribute the nitrate formation to homogeneous reaction based on the fit of NO3/SO4 and NH4/SO4 molar ratios. Then, for night time, you conclude that aqueous reactions dominate nitrate formation based on the increasing trend of NO3 with ALWC. Then you

justify not considering the fitting approach used for daytime based on the fact that NOx and SO2 emissions decreased and NH3 emissions increased. Is the RH high throughout night time? Couldn't it mean that you have both homogeneous and heterogeneous reactions occurring? If you can show that "HNO3 was firstly heterogeneously formed through the hydrolysis of N2O5, then excess NH3 was uptake by wet particles and neutralised HNO3 forming ammonium nitrate" dominate nitrate formation at night, then I would simply not mention the night time NO3/SO4 vs NH4/SO4 fitting. I suggest mentioning why it is not applicable first, and then talk about the aqueous reactivity because this could lead the reader to doubt the fitting relevance during daytime as well.

The regional transport discussion should be moved prior to the "Case studies for haze pollution" section as the content does not provide specific details or information that contribute to a better understanding of the haze episodes. And "Evolution of chemical composition during fog periods" would probably correspond more as the second subsection of "Case studies for haze pollution".

Reply: Thanks for the reviewer's good suggestion. As the reviewer mentions, the discussion of $[NO_3^-/SO_4^{2-}]$ vs. $[NH_4^+/SO_4^{2-}]$ fitting during night time indeed makes the fitting relevance during daytime doubtful. We have reconstructed the discussion of this part in the revised manuscript as the reviewer suggests, and added the part of discussion on the probable nocturnal nitrate formation pathways according to previous studies. (lines 273-292)

The intercept of the regression line on the $[NH_4^+]/[SO_4^{2-}]$ axis was 1.56, which was close to 1.5, as suggested by Pathak et al. (2009), implying that the nitrate formation was mainly driven by the homogeneous reactions (Sun et al., 2011). Indeed, the nitrate concentration and nitrogen oxidation ratio (NOR = $n(NO_3^-)/[n(NO_2) + n(NO_3^-)]$) increased as the Ox concentration increased (as shown in Fig. 4), and exhibited a strong $O_3/Ox$ ratio dependency, which further demonstrated the homogeneous daytime formation of nitrate.

The emission of NOx and SO$_2$ had been reduced while NH$_3$ increased in the past almost ten years, which resulted in the ammonium-rich condition in the atmosphere (Fu et al., 2017; Liu et al., 2018). Despite this, a recent study showed that the nocturnal nitrate formation was not sensitive to NH$_3$, and even increased slightly as NH$_3$ decreased, which was likely due to the aerosol acidity effects on the partitioning of nitrate (Wen et al., 2018). Thus, the fitting of $[NO_3^-]/[SO_4^{2-}]$ vs. $[NH_4^+]/[SO_4^{2-}]$ might not be applicable for identifying the nitrate formation process during

nighttime. The average $O_3$ concentration was 13.7 μg/m$^3$ and the average RH was 83.3 % during nighttime, which favoured the aqueous-phase reactions to occur. Higher nitrate concentration was observed with increasing ALWC during nighttime (as illustrated in Fig. S2), and so was NOR. This phenomenon further implies the heterogeneous hydrolysis of $N_2O_5$ might dominate the formation of nocturnal nitrate. The results were consistent with the study of Tian et al. (2019), which showed that heterogeneous hydrolysis of $N_2O_5$ dominated nitrate formation during nighttime, while photochemical reactions also played an important role in nitrate formation during daytime in two megacities in Sichuan Basin.

As the reviewer suggests, we have put the discussion of regional transport before "Case studies for haze pollution" section. Also, we have put the "Evolution of chemical composition during fog periods" section as a subsection of "Case studies for haze pollution" in the revised manuscript.

Minor comments on manuscript:

Page 1 lines 21-23: "The fine aerosol chemical composition was characterised by using a time-of-flight aerosol chemical speciation monitor (ToF-ACSM) with the aim of inorganic and organic aerosol characterisation and source apportionment." Please, rephrase.

Reply: Thanks for the reviewer's careful check, we have rephrased this sentence in the revised manuscript. (lines 21-23)

The fine aerosol chemical composition was characterised by using a time-of-flight aerosol chemical speciation monitor (ToF-ACSM), which also provided detailed information on the sources for organic aerosols (OA).

Page 1 line 25: Please choose a more appropriate word than "occupied"

Reply: We have replaced this word with "took up" in the manuscript. (line 25)

The average concentration of non-refractory fine particles (NR-PM$_{2.5}$) was 98.5 ± 38.7 μg/m$^3$, and organics aerosols (OA), nitrate, sulphate, ammonium, and chloride **took up** 40.3, 28.8, 10.6, 15.3 and 5.1 % of PM$_{2.5}$.

Page 3 lines 64-66: "The emission of SO2 had been reduced dramatically over the past ten years in China; however, NOx did not show a significant reduction." Please add references to support these

trends.

Reply: Thanks for the reviewer's careful check. Relevant references have been added in the revised manuscript. (lines 65-66)

The emission of $SO_2$ had been reduced dramatically over the past ten years in China; however, NOx did not show a significant reduction (Liu et al., 2019; Zhou et al., 2021).

Reference

Liu, M., Huang, X., Song, Y., Tang, J., Cao, J., Zhang, X., Zhang, Q., Wang, S., Xu, T., Kang, L., Cai, X., Zhang, H., Yang, F., Wang, H., Yu, J. Z., Lau, A. K. H., He, L., Huang, X., Duan, L., Ding, A., Xue, L., Gao, J., Liu, B.Zhu, T., 2019. Ammonia emission control in China would mitigate haze pollution and nitrogen deposition, but worsen acid rain. Proceedings of the National Academy of Sciences 116 (16), 7760-7765.

Zhou, W., Chen, C., Lei, L., Fu, P.Sun, Y., 2021. Temporal variations and spatial distributions of gaseous and particulate air pollutants and their health risks during 2015–2019 in China. Environ. Pollut. 272, 116031.

Page 3 line 68: "Compared to SIA, the formation process of SOA was more complicated (Chen et al., 2017)." Which formation process are you referring to? Or are you referring to SOA formation in general and therefore it includes multiple processes/pathways… As described later in the paragraph.

Reply: Sorry that we did not make this point clear. The "formation process" mentioned here refers to the multiple processes/pathways described in this paragraph.

Page 4 line 82: "was also suffering severe haze pollution", if it is still happening, I would use present continuous tense.

Reply: Thanks for the reviewer's suggestion. The haze pollution is still a severe atmospheric problem in SCB, especially during winter. According to the report of 'Sichuan Ecology and Environment Statement 2021', the annual average $PM_{2.5}$ concentration of Chengdu was ~40 μg/m$^3$, which exceeded the grade Ⅱ of National Ambient Air Quality Standard (AAQS, GB 3095-2012) (30

μg/m³). We have changed the tense in the manuscript as the reviewer suggested. (line 83-85)

the Chengdu-Chongqing city cluster, located in the Sichuan Basin (SCB) in Southwest China, is also suffering severe haze pollution (Tao et al., 2017; Tan et al., 2019)

Page 5 line 148: "~84 cc/min" for consistency with previous flow (line 141) you should either write the equivalent value in L/min

Reply: Thanks for the reviewer's suggestion. We have changed the flow rate of "~84 cc/min" to "~0.084 L/min" in the revised manuscript. (line 150)

Briefly, a 100 μm critical orifice and an aerodynamic lens were settled in the front inlet system to focus the ambient particles into a concentrated and narrow beam with a flow rate of ~0.084 L /min.

Page 7 – Data Process section: information on the elemental analysis with the TOF-ACSM is lacking.

Reply: We have added the information of the elemental analysis in the Data Process section in the revised manuscript. (lines 175-179)

The triangle plot of f44 (ratio of m/z 44 to total signal in the component mass spectrum) versus f43 (ratio of m/z 43 to total signal in the component mass spectrum) and f44 versus f60 (ratio of m/z 60 to total signal in the component mass spectrum) were applied to characterise the evolution of OA (Ng et al., 2010). The O/C and H/C were determined by the parameterization proposed by Canagaratna et al. (2015).

Reference

Ng, N. L., Canagaratna, M. R., Zhang, Q., Jimenez, J. L., Tian, J., Ulbrich, I. M., Kroll, J. H., Docherty, K. S., Chhabra, P. S., Bahreini, R., Murphy, S. M., Seinfeld, J. H., Hildebrandt, L., Donahue, N. M., DeCarlo, P. F., Lanz, V. A., Prévôt, A. S. H., Dinar, E., Rudich, Y., and Worsnop, D. R.: Organic aerosol components observed in Northern Hemispheric datasets from Aerosol Mass Spectrometry, Atmos. Chem. Phys., 10, 4625-4641, https://doi.org/10.5194/acp-10-4625-2010, 2010.

Canagaratna, M. R., Jimenez, J. L., Kroll, J. H., Chen, Q., Kessler, S. H., Massoli, P., Hildebrandt Ruiz, L., Fortner, E., Williams, L. R., Wilson, K. R., Surratt, J. D., Donahue, N. M., Jayne, J. T., and

Worsnop, D. R.: Elemental ratio measurements of organic compounds using aerosol mass spectrometry: characterization, improved calibration, and implications, Atmos. Chem. Phys., 15, 253-272, https://doi.org/ 10.5194/acp-15-253-2015, 2015.

Page 10 line 7: "planet boundary (PBL) height" I assume the "layer" is missing in that sentence.

Reply: Thanks for the reviewer's check. Indeed, we left out "layer" in this sentence. We have deleted the discussion of the diurnal variation of gaseous pollutants and PM2.5 compositions in the revised manuscript as the other reviewer suggests. We have corrected it in other paragraphs in the revised manuscript. (lines 498-499)

Except for OOA and nitrate, all species kept decreasing during post-fog periods, which might be attributed to the increase of the planet boundary layer (PBL) height.

Page 10 line 261: You mention biomass burning as a source of Chloride. Any idea of fuel used or burning conditions?

Reply: The north and west of the observation site are croplands and villages. Crop residue (such as rice, wheat, rape and corn straws), wood with branches and leaves are commonly used as fuels for cooking and heating in the nearby villages. These biofuels are usually burned (flaming) in a cookstove and/or in a washbasin for cooking and heating, respectively (as shown in the figure below). We often saw the smoke of biomass burning in the morning and evening at the observation site.

[Figure]

Figure 1 Typical biomass burning for cooking in a cookstove in the villages.

Page 11 line 275: "If [NO3-]/[ SO42-] linearly correlated with [NO3-]/[SO42-] under ammonium-rich conditions", shouldn't it be linearly correlated with "[NH4+]/[ SO42-]"?

Reply: Thanks for the reviewer's careful check. We have corrected it in the revised manuscript. (lines 259-260)

If $[NO_3^-]/[SO_4^{2-}]$ linearly correlated with $[NH_4^+]/[SO_4^{2-}]$ under ammonium-rich conditions $([NO_3^-]/[SO_4^{2-}] \geq 1.5)$

Page 13 line 348: You mentioned that chloride is a biomass burning tracers and that these BB could be related to cooking and heating. Which of these sources would emit Cl-?

Reply: As discussed above, crop residues and wood with branches and leaves are commonly used for cooking and heating. Previous studies showed that wheat and rape straws burning (flaming) had high $Cl^-$ emission factors (Hays et al., 2005; Engling et al., 2009), and the emission of these biofuels burning indeed contributed to high concentration $Cl^-$ in SCB during the BB period (Tao et al., 2013). Although wood burning produced $Cl^-$, the amount of $Cl^-$ emitted from wood burning was not as large as rice and wheat straws (Alves et al., 2011). Thus, cooking would contribute more $Cl^-$ compared to heating, especially in the morning.

Reference

Alves, C., Gonçalves, C., Fernandes, A. P., Tarelho, L., and Pio, C.: Fireplace and woodstove fine particle emissions from combustion of western Mediterranean wood types, Atmos. Res., 101, 692-700, https://doi.org/10.1016/j.atmosres.2011.04.015, 2011.

Engling, G., Lee, J. J., Tsai, Y., Lung, S. C., Chou, C. C. K., Chan, C., and Chan, K. C. C.: Size-Resolved Anhydrosugar Composition in Smoke Aerosol from Controlled Field Burning of Rice Straw, Aerosol Sci. Tech., 43, 662-672, https://doi.org/10.1080/02786820902825113, 2009.

Hays, M. D., Fine, P. M., Geron, C. D., Kleeman, M. J., and Gullett, B. K.: Open burning of agricultural biomass: Physical and chemical properties of particle-phase emissions, Atmos. Environ., 39, 6747-6764, https://doi.org/10.1016/j.atmosenv.2005.07.072, 2005.

Tao, J., Zhang, L., Engling, G., Zhang, R., Yang, Y., Cao, J., Zhu, C., Wang, Q., and Luo, L.: Chemical composition of PM2.5 in an urban environment in Chengdu, China: Importance of springtime dust storms and biomass burning, Atmos. Res., 122, 270-283, https://doi.org/10.1016/j.atmosres.2012.11.004, 2013.

Page 14 lines 369-373: "The average OOA concentration did not change significantly with increasing ALWC during daytime, suggesting the less contribution of aqueous state reaction to the formation of OOA. During nighttime, the average OOA concentration showed an increasing trend when ALWC < 200 μg/m3 and kept relatively constant subsequently, suggesting the aqueous-phase reactions did not significantly affect the formation of OOA" You can maybe shorten this part by saying that aqueous reactions are not significant pathway toward OOA formation during day- and night-time.

Reply: Thanks for the reviewer's suggestion. We have shortened this part in the revised manuscript. (lines 345-349)

Except that the average OOA concentration showed an increasing trend when ALWC < 200 μg/m$^3$ during nighttime, OOA concentration did not change significantly with increasing ALWC during both day- and nighttime, suggesting the aqueous-phase reactions were not significant pathway toward OOA formation.

Page 15 lines 403-405: "Higher RH was observed for those data points within the region of aged BBOA in the f44 vs. f60 space". Although, I agree that BBOA oxidation probably occurs in the aqueous phase, in Figure 8, it seems that the RH is high for most of the points falling in the f44 vs f60 triangle, except for the data with f44 > 0.15 and 0.08>f60>0.05, where the RH seems lower. Also is there a reason behind using RH here instead of ALWC as used in the previous comparison?

Reply: It is a good question. We take the region of aged BBOA in the $f$44 vs. $f$60 reported by Paglione et al. (2020) as reference (as shown in the figure below). It can be found that most of the data points (including those with $f$44 > 0.15 and 0.08 > $f$60 > 0.05 as the reviewer mentions) in the present study are within this region and correspond to high RH conditions. It seems that our description might confuse the readers. We have made the description clearer in the revised manuscript. (lines 378 -

[Figure]

Figure 1 Triangle plots of *f*44 vs. *f*60 (a) reported by Paglione et al. (2020) and (b) in the present study. The ranges of f44 and f60 for 'Aged BBOA' and 'Fresh BBOA' are marked in (a).

The main reason why the data points in Figure 8(d) are mapped with RH instead of ALWC is that the ALWC simulated by ISORRIPIA increased dramatically for the conditions with RH > 95 %. The ALWC simulated for the conditions with RH > 95 % are higher 2-3 orders of those with RH ranging from 60-90 %. It is not obvious to observe the effect of ALWC on OOA if mapping the data points with ALWC directly, so we used RH to map the data points. However, the reviewer's suggestion reminds us that we can use the logarithm of ALWC to map the data points. We have changed lg(ALWC) with RH in figure 8(d) in the revised manuscript.

Compared to the effects of Ox, the increasing ALWC did not seem to push *f*60 to the left upper region. Most of the data points, which corresponded to high ALWC, were within the region of aged BBOA in the f44 vs. f60 space as defined previously by Paglione et al. (2020), indicating the probable aqueous-phase oxidation of BBOA.

[Figure]

Fig. 8 Triangle plots of (a), (c) f44 (ratio of m/z 44 to total signal in the component mass spectrum) vs. f43 (ratio of m/z 43 to total signal in the component mass spectrum), and (b), (d) f44 vs. f60 (ratio of m/z 60 to total signal in the component mass spectrum) during the whole campaign. The colour scale in (a) and (b) represents Ox concentration, and that in (c) and (d) represents lg(ALWC). The solid lines in (a) and (c) are derived from the results reported by (Ng et al., 2010). The dashed line representing the background value of secondary aged OA and the solid guidelines in (b) and (d) are derived from (Cubison et al., 2011). The f44 vs. f43 and f44 vs. f60 for different OA sources reported in previous studies are also shown (Kim et al., 2019; Ng et al., 2011; Paglione et al., 2020; Wang et al., 2016; Zhao et al., 2019).

Page 16 line 428: change "Table S2" to Table S3.

Reply: Corrected. (line 431)

The synoptic conditions and aerosol chemical composition for each haze episode were summarised in Table. S3.

Page 17 line 460 and after: as a cluster represent several "air masses", the plural form is probably

more adapted, especially that you use "air parcels" later on in the paragraph.

Reply: Thanks for the reviewer's suggestion. We have replaced "air mass" with "air parcels" throughout the revised manuscript.

Page 18 PSCF discussion: more details about the threshold value used could be added in the text or in Figure 12.

Reply: Thanks for the reviewer's suggestion. We have added the details of the threshold values used in Figure 12 in the revised manuscript. (lines 642-658)

[Figure]

Fig. 10 Simulation results of PSCF for (a) organics, (b) nitrate, (c) sulphate, (d) HOA, (e) BBOA, and (f) OOA during the whole campaign. The 50th percentile of the concentrations for each composition (organics: 39.5 μg/m$^3$, nitrate: 27.8 μg/m$^3$, sulphate: 9.5 μg/m$^3$, HOA: 7.6 μg/m$^3$, BBOA: 8.7 μg/m$^3$, OOA: 15.2 μg/m$^3$) were used as thresholds in the PSCF analysis. The areas of Deyang and Sichuan Province are marked in (a).

Page 20 lines 534-540: "The average elemental O:C showed an increasing trend from pre-fog periods to post-fog/foggy periods, while H:C did not change significantly for different fog events, suggesting the OA became more oxidised. As shown in Fig. S6, the mass fractions of OOA increased, while the contribution of BBOA and HOA decreased from pre-fog periods to post-fog/foggy periods for the three fog events. As a consequence, the O:C increased in line with the increased contribution of OOA." The O:C and H:C could be added to Figure S6

Reply: Thanks for the reviewer's suggestion. The O:C and H:C have been added in Figure S6 in the revised supplementary information.

[Figure]

Fig. S9 Variation of O:C of OA and relative contribution of OOA, BBOA, and HOA to OA during the evolution of (a), (d) F1, (b), (e) F2, and (c), (f) F3.

Figure 2 would benefit from a different (perhaps lighter) background as the yellow makes it difficult to distinguish between SO4, NH4 and Chl.

Reply: Thanks for the reviewer's suggestion. The colour of the background has been replaced with a lighter one.

[Figure]

Fig. 2 Time series of (a) relative humidity (RH) and temperature (T); (b) wind direction and wind

speed; (c), (d) CO, $NO_2$, $SO_2$, and $O_3$ mass concentrations and solar radiation; (e) mass concentration of organics, nitrate, sulphate, ammonium, and chloride in NR-PM$_{2.5}$. The yellow-shaded areas represent the intervals of H1, H2, and H3, respectively. The light blue-shaded areas represent the intervals of F1, F2, and F3, respectively.

Figure 3: As you discuss day/night time nitrate formation and the effect of RH at night, could you perhaps add RH diurnal variation. Or a figure with the diurnal cycles of meteorological parameters and PBL could be added in the supplement.

Reply: Thanks for the reviewer's suggestion. Since the discussions of diurnal variations of PM$_{2.5}$ compositions and meteorological parameters have been deleted in the revised manuscript. We put the diurnal cycles of PM$_{2.5}$ chemical compositions, gaseous pollutants, meteorological parameters and PBL together in the supplementary information.

[Figure]

Fig. S3 Diurnal variations of (a) chemical composition in NR-PM$_{2.5}$, (b) $O_3$, $NO_2$, and $SO_2$, (c) RH and temperature, (d) planet boundary layer height (PBLH) and solar radiation (SR). The PBLH was derived from the European Centre for Medium-Range Weather Forecasts (ECMWF) dataset of ERA5 hourly data (https://cds.climate.copernicus.eu/cdsapp#!/home).

Figure 6: it would be helpful to add some background to evidence the fog periods on the time series of the OA sources.

Reply: Thanks for the reviewer's suggestion. We have added backgrounds to mark fog periods.

[Figure]

Fig. 5 Mass spectrum of (a) HOA, (b) BBOA, and (c) OOA resolved by PMF. The time series of each OA source and corresponding tracers are depicted in the right panel. The light blue shaded areas represent the intervals of foggy periods.

Minor comments on supplement:

Page 3 Figure S2: It may be easier to use a lighter blue for nitrate as the mean is hard to distinguish. Could the dataset be separated between day/night time as it supports the discussion between secondary inorganic aerosol day/night formation?

Reply: Thanks for the reviewer's suggestion. We have changed the colour of nitrate to a lighter blue in the relevant figures. Also, we have separated the dataset into day- and nighttime ones to show the difference of SIA formation during day- and nighttime.

[Figure]

Fig. S1 Variation of nitrate and sulphate as Ox/ALWC increases during (a) daytime (left column) and (b) nighttime (right column). The data of nitrate and sulphate concentrations were grouped into different bins according to 20 μg/m³ increment of Ox, and 100 μg/m³ increment of ALWC. The mean (square), 50th (horizontal line inside the box), 25th and 75th percentiles (lower and upper box), and 10th and 90th percentiles (lower and upper whiskers) of the box chart are marked in (a).

Reviwer2

The authors reported measurement results of PM2.5 components at a site in Sichuan basin, China, using a time-of-flight aerosol chemical speciation monitor (ToF-ACSM). General results of the one-month campaign in winter 2021/2022 were presented with routine but rigorous data analysis tools. Three haze events, each accompanied with a foggy period, were selected for case studies to identify the reasons behind haze formation. The authors concluded that intensive biomass burning and rapid nitrate formation might be the reason behind the formation of those haze events. The study is in general well designed and properly conducted, and the manuscript is fairly well written. I therefore recommend Minor Revision before publication.

Reply: Thanks for the reviewer's positive comment. We hope that the results of the present study will improve our knowledge of the factors driving haze formation in SCB.

Main:

The authors tried to make a point in the title that "intensive" biomass burning and "rapid" formation "drive" severe haze formation in their campaign. Yet, I do not see clear evidence supporting such a statement. First, for biomass burning, BBOA contributed 20-30% to OA, and maybe 10-15% of NR-PM2.5 during haze events (Figure 10a). Yes, it is non-negligible, but I would not say that it drives the haze formation. In addition, I do not see evidence for "intensive" biomass burning during haze events. Maybe showing some fire spot data from satellite archive will help. Second, for nitrate, the contribution of around 30% to NR-PM2.5 during haze events is of course quite substantial. But I do not see any evidence of "rapid" formation of nitrate. Maybe showing some cases of fast growing of nitrate concentrations in some haze events would help.

Reply: Thanks for the reviewer's suggestion. We have added the discussion of satellite observation results showing fire spots during the haze episodes in the manuscript. (lines 465 -467)

The fire maps (as illustrated in Fig. S8) showed that more fire spots during H2 and H3 were observed around Deyang compared to non-haze episodes, suggesting the biomass burning activities were more intensive during these haze episodes.

[Figure]

Fig. S8 Fire maps of areas around Deyang during (a) non-haze, (b) H2 and H3 periods. The Fire Maps were acquired from Fire Information for Resource Management System (FIRMS) developed by the National Aeronautics and Space Administration (NASA). The data of VIIRS (375m) was used (https://firms.modaps.eosdis.nasa.gov/active_fire/).

Also, as the reviewer suggested, we have added the growth rate of nitrate to support the fast nitrate formation during the evolution of haze pollution. (lines 450-457)

The average $NO_3^-$ formation rate as a function of $PM_{2.5}$ concentration during H1 was depicted in Fig. S7. The $NO_3^-$ formation rate increased fast as $PM_{2.5}$ concentration increased from 50 to 110 $\mu g/m^3$, which also showed the rapid formation of nitrate contributed to haze formation. In contrast, the average nitrate formation rates were below zero when the $PM_{2.5}$ concentration was $< 130$ $\mu g/m^3$ during H2 and H3, suggesting nitrate formation did not play an important role at the early stage of H2 and H3. Although the nitrate formation rate decreased when $PM_{2.5}$ concentration was $> 110$ $\mu g/m^3$ during H1, it remained positive, suggesting the nitrate concentration increased gradually.

[Figure]

Fig. S7 Average nitrate formation rate as a function of PM$_{2.5}$ concentration during H1, H2 and H3

Sections 3.1 – 3.3 are quite routine and do not contribute much to the value of this study. I suggest shortening these three sections and focus on (expanding) discussion of the reasons behind haze formation (i.e., section 4).

Reply: Thanks for the reviewer's suggestion. As the reviewer suggests, we have cut some discussions on the diurnal variations of gaseous pollutants and PM2.5 chemical compositions. We have also shortened the discussion on nitrate formation during nighttime. Besides, we have added some discussions on the nitrate formation rate and fire spots to support the result of the rapid nitrate formation and biomass burning in section 3.5 as the reviewer suggests.

There are a few contradictory statements in the manuscript that I suggest the authors to resolve in the revision. For instance, it was suggested that aqueous-phase reaction was not important in OOA formation (L557), but in the discussion in L511 the authors suggested otherwise; the discussion on nitrate formation (L309-316) is interesting, but I do not follow 1) why the abundant ammonia can accommodate plenty of basic species (L310), and 2) how did the authors reach the conclusion that nitric acid was formed heterogeneously (which the authors thought that was not important in L290 and L303), and then take up ammonia?

Reply: Thanks for the reviewer's questions. Except that the average OOA concentration showed an increasing trend when ALWC < 200 μg/m$^3$ during nighttime, OOA concentration did not change significantly with increasing ALWC during both daytime and nighttime, suggesting the aqueousphase reactions were not significant pathway promoting OOA formation. In spite of this, it did not mean that aqueous-phase reactions did not occur during foggy periods. In fact, previous studies showed that SOA could be formed through aqueous-phase reactions under high RH conditions (Kuang et al., 2020; Duan et al., 2021). Thus, we thought that OOA could be formed through aqueous-phase reactions during foggy period, and offset the scavenging effect of fog droplets.

For the second and third questions, our description might be confusing and not appropriate. Since the atmosphere was under ammonium-rich conditions during nighttime, we deduced that wet particles would uptake $NH_3$ and neutralise $HNO_3$ subsequently, thus generating ammonium nitrate. However, the simulation results of Wen et al. (2018) showed that the $NH_3$ in excess would decrease the aerosol acidity and allow the reaction of $NO_2^+$ with $Cl^-$ to happen during nighttime, hence restricting the formation of nitrate. Thus, the nitrate formation was primarily formed via the heterogeneous hydrolysis of $N_2O_5$, instead of the neutralisation between $HNO_3$ (aq, s) and $NH_3$ (g). As the other reviewer suggests, we have reconstructed this part in the revised manuscript. (lines 279-289)

The emission of NOx and $SO_2$ had been reduced while $NH_3$ increased in the past almost ten years, which resulted in the ammonium-rich condition in the atmosphere (Fu et al., 2017; Liu et al., 2018). In spite of this, a recent study showed that the nocturnal nitrate formation was not sensitive to $NH_3$, and even increased slightly as $NH_3$ decreased, which was likely due to the aerosol acidity effects on the partitioning of the nitrate formation (Wen et al., 2018). Thus, the fitting of $[NO_3^-]/[SO_4^{2-}]$ vs. $[NH_4^+]/[SO_4^{2-}]$ might not be applicable for identifying the nitrate formation process during nighttime. The average $O_3$ concentration was 13.7 $\mu g/m^3$ and average RH was 83.3 % during nighttime, which favoured the aqueous-phase reactions to occur. Higher nitrate concentration was observed with increasing ALWC during nighttime (as illustrated in Fig. S2), and so was NOR. This phenomenon further demonstrated the heterogeneous hydrolysis of $N_2O_5$ might dominate the formation of nocturnal nitrate.

Reference

Duan, J., Huang, R., Gu, Y., Lin, C., Zhong, H., Wang, Y., Yuan, W., Ni, H., Yang, L., Chen, Y., Worsnop, D. R., and O'Dowd, C.: The formation and evolution of secondary organic aerosol during summer in Xi'an: Aqueous phase processing in fog-rain days, Sci. Total Environ., 756, 144077,

https://doi.org/10.1016/j.scitotenv.2020.144077, 2021.

Kuang, Y., He, Y., Xu, W., Yuan, B., Zhang, G., Ma, Z., Wu, C., Wang, C., Wang, S., Zhang, S., Tao, J., Ma, N., Su, H., Cheng, Y., Shao, M., and Sun, Y.: Photochemical Aqueous-Phase Reactions Induce Rapid Daytime Formation of Oxygenated Organic Aerosol on the North China Plain, Environ. Sci. Technol., 54, 3849-3860, https://doi.org/10.1021/acs.est.9b06836, 2020.

Wen, L., Xue, L., Wang, X., Xu, C., Chen, T., Yang, L., Wang, T., Zhang, Q., and Wang, W.: Summertime fine particulate nitrate pollution in the North China Plain: increasing trends, formation mechanisms and implications for control policy, Atmos. Chem. Phys., 18, 11261-11275, https://doi.org/10.5194/acp-18-11261-2018, 2018.

Minor:

L30: add "processes" after "aqueous-phase"?

Reply: Thanks for the reviewer's careful check. We have added "processes" after "aqueous-phase" in the revised manuscript. (line 30)

Nitrate formation was promoted by gas-phase and aqueous-phase oxidation, while sulphate was mainly formed through aqueous-phase process.

L61 and a few other places: citation format not in accordance with that of ACP.

Reply: Thanks for the reviewer's careful check. We have corrected the citation format in accordance with the requirement of ACP in the revised manuscript.

L387: aqueous-state should be aqueous-phase?

Reply: Corrected. (lines 347-349)

OOA concentration did not change significantly with increasing ALWC during both daytime and nighttime, suggesting the aqueous-phase reactions were not a significant pathway toward OOA formation.

Figure 12: better to clearly indicate the site, and Deyang and Sichuan in the maps. It is hard to follow when they are referred to in L475-485.

Reply: Thanks for the reviewer's suggestion. We have adjusted the scale of the map in figure 12 and made it clear to distinguish Deyang and Sichuan

[Figure]

Fig. 10 Simulation results of PSCF for (a) organics, (b) nitrate, (c) sulphate, (d) HOA, (e) BBOA, and (f) OOA during the whole campaign. The 50th percentile of the concentrations for each composition (organics: 39.5 µg/m³, nitrate: 27.8 µg/m³, sulphate: 9.5 µg/m³, HOA: 7.6 µg/m³, BBOA: 8.7 µg/m³, OOA: 15.2 µg/m³) were used as thresholds in the PSCF analysis. The areas of Deyang and Sichuan Province are marked in (a).

---

## Author Response (AR2)

Manuscript No. ACP-2022-477

Tittle: Measurement report: Intensive biomass burning emissions and rapid nitrate formation drive severe haze formation in Sichuan basin, China: insights from aerosol mass spectrometry

The authors gratefully thank all the reviewers for their comments and suggestions. We have revised our manuscript according to the two reviewers' suggestions and comments. **All the changes and responses to the reviewers' comments are listed below point-by-point. The changes are highlighted with red in the revised manuscript.** We sincerely hope this manuscript will be acceptable for publication in *Atmospheric Chemistry and Physics*.

Minor comments:

Page 6 - line 143 : "the flow rate was maintained at 3 L/min with a flow meter" I assume the author meant a mass flow controller instead of a flow meter to maintain the 3LPM flow rate.

Reply: Thanks for the reviewer's suggestion. Our flow meter has the function of controlling flow rate. Despite this, as the reviewer suggests, it is more precise to use 'flow controller' in the manuscript. We have changed the 'flow meter' to 'flow controller' in the revised manuscript. (line 143)

the flow rate was maintained at 3 L/min with a mass flow controller

Page 6 – line 145: "then was dried by a Nafion drier" was the RH below 30-40%? The author could maybe indicate a range of RH.

Reply: Sorry for missing the information of the RH after the Nafion drier. Except for the fog periods, the RH of the air samples dried by the Nafion drier ranged 35-46 % during the whole campaign. The RH of the air samples dried by the Nafion drier during fog periods could reach ~56 %. We have added the RH of the air samples dried by the Nafion drier in the revised manuscript. (lines 145-147)

the ambient air would go through a PM2.5 cyclone (URG-2000-30ED, USA) to remove coarse particles, then was dried by a Nafion drier. The relative humidity of the air samples dried by the Nafion drier usually ranged from ~35% to 46 %, and could reach ~56 % during fog events.

Page 6 line 151: "which made the PM2.5 measurement available". I suggest to rephrase this part,

e.g. "the PM2.5 allows to chemically characterize the PM2.5 fraction" or something similar.

Reply: Thanks for the reviewer's suggestion. We have rephrased this part as the reviewer suggests. (lines 152-153)

It should be mentioned that a PM2.5 lens was used during the whole campaign, which allowed to chemically characterise the PM2.5 composition (Xu et al., 2017).

Page 9 – lines 218-219: "an arrival height of 500 m which is above ground level (AGL) for target analysis in the HYSPLIT model to diminish the effects of surface friction (Polissar et al., 2001) this height value and greater are regarded as in the open height of the planetary boundary layer in winter and are more useful for long-range transport". Based on the PBL height reported in Figure S3d this height is most of the time above the PBL observed at the sampling site? Would using a fraction of the BL in HYSPLIT result in similar results/be more appropriate? Maybe you try to justify the height choice in the second part of this sentence but it doesn't sound very conclusive, especially when you mentioned earlier in the manuscript that the landscape in the SCB basin "is unfavourable for either horizontal transport or vertical diffusion".

Reply: The height of 500 m above ground level (AGL) exceeded the PBLH most of the time, as shown in the figure 1 below. As a fact, we had compared the simulation results for the arrival heights of 200 m, 500 m and 1000 m AGL (as shown in the figure 2 below). Although the appropriate number of mean trajectories for the arrival height of 1000 m was three, the origins and the transport paths of the cluster-mean trajectories for these three heights were similar. The air parcels from the east and northeast of Sichuan Province took up major parts in all air parcels.

The HYSPLIT model computation is a terrain following calculation, and the trajectories can never go below ground. The terrain issue of valley/basin can be ignored if the valley/basin is relatively wide and large, and the arrival height can be set to a value of interest (one is referred to https://www.arl.noaa.gov/hysplit-frequenctly-asked-questions for more detail information). Since the 500 m height covers the height above and below PBLH, we chose this height to achieve an overall consideration of both long-range transport and local transport. Tao et al. (2013) also used this height to analyse the transport paths in Sichuan Basin. Wang et al., (2018) even used a lower arrival height (300 m) in their study to identify major transport paths in two megacities in Sichuan Basin. As the reviewer suggests, the description 'this height value and greater are regarded as in the

open height of the planetary boundary layer in winter and are more useful for long-range transport' is somewhat not conclusive. We have deleted this sentence in the revised manuscript.

[Figure]

Figure 1 Temporal profile of planet boundary layer height (PBLH) during the campaign. The red line shows the 500 m height. The PBLH is derived from the European Centre for Medium-Range Weather Forecasts (ECMWF) dataset of ERA5 hourly data (https://cds.climate.copernicus.eu/cdsapp#!/home)

[Figure]

Figure 2 Simulation results of 48 h backward air parcels cluster-mean trajectories for arrival heights of (a) 500 m, (b) 200 m and (c) 1000 m during the campaign.

Reference

Tao, J., Zhang, L., Engling, G., Zhang, R., Yang, Y., Cao, J., Zhu, C., Wang, Q., and Luo, L.: Chemical composition of PM2.5 in an urban environment in Chengdu, China: Importance of springtime dust storms and biomass burning, Atmos. Res., 122, 270-283, https://doi.org/10.1016/j.atmosres.2012.11.004, 2013.

Wang, H., Tian, M., Chen, Y., Shi, G., Liu, Y., Yang, F., Zhang, L., Deng, L., Yu, J., Peng, C., and

Cao, X.: Seasonal characteristics, formation mechanisms and source origins of PM2.5 in two megacities in Sichuan Basin, China, Atmos. Chem. Phys., 18, 865-881, https://doi.org/10.5194/acp-18-865-2018, 2018.

Page 15 line 403: "Clsuter2"      cluster 2

Reply: Corrected. (line 403)

Page 17 line 462: "the rest composition did not change significantly" could you rephrase.

Reply: Thanks for the reviewer's suggestion. We have rephrased this part. (lines 461-463)

During H3, the fractions of nitrate and BBOA in $PM_{2.5}$ increased, while OOA decreased and HOA, sulphate, ammonium, chloride did not change significantly as the $PM_{2.5}$ concentration increased.

Page 18 line 485: "due to the aqueous-phase reaction"      "due to aqueous-phase reactions"

Reply: Corrected (lines 484-485)

The domination of secondary species in $PM_{2.5}$ during F1 was probably due to the aqueous-phase reactions

Page 19 line 502: "Distinguished from H1"      rephrase

Reply: Thanks for the reviewer's suggestion. We have rephrased this part. (lines 502-503)

All species (except for HOA) increased during the foggy period from the pre-fog period during F2, which was different from the case during F1.

Page 19 lines 509-511: "The increase of BBOA in the present study was attributed to the intense emission from biomass burning during the foggy period, which overwhelmed the scavenging effects of fog droplet" Could it be that biomass burning VOCs partitioning is enhanced by the high RH?

Reply: This is a good question. As reported by a recent study by Xiao et al. (2022), biomass burning emits large quantities of phenols, which readily partition into the atmospheric aqueous phase and subsequently may react to produce aqueous secondary organic aerosol (aqSOA). At first, we also thought that the increase of BBOA could be attributed to the enhanced partitioning of biomass burning VOCs. However, the increase of BBOA (compared to pre-fog period) was only observed

during the foggy period of F2. If it was the case that high RH enhanced the partitioning of biomass burning VOCs, an increase of BBOA should be also observed during the foggy periods of F1 and F3. In fact, compared to pre-fog periods, BBOA decreased during the foggy periods of F1 and F3. Thus, we thought it more reasonable that the increase of BBOA during the foggy period of F2. However, as the reviewer mentions, it might be more comprehensive to add some discussions on the contribution of biomass burning VOCs partitioning. We have added relevant discussions in the revised manuscript. (lines 509-514)

The increase of BBOA in the present study was likely attributed to the intense emission from biomass burning during the foggy period, which overwhelmed the scavenging effects of fog droplets. The enhanced partitioning of biomass burning VOCs under high RH conditions might also contribute the increase of BBOA. For example, a recent study showed that the large quantities of phenols from biomass burning emission would readily partition into the atmospheric aqueous phase (Xiao et al., 2022).

Reference

Xiao, Y., Hu, M., Li, X., Zong, T., Xu, N., Hu, S., Zeng, L., Chen, S., Song, Y., Guo, S., and Wu, Z.: Aqueous secondary organic aerosol formation attributed to phenols from biomass burning, Sci. Total Environ., 847, 157582, https://doi.org/10.1016/j.scitotenv.2022.157582, 2022.

Page 19 lines 522-523: "the mass fractions of OOA increased, while the contribution of BBOA and HOA decreased from pre-fog periods to post-fog/foggy periods for the three fog events." Scavenging is mentioned as the main reason for the decrease in HOA and BBOA. Would it be possible that BBOA and HOA would just get oxidized during the fog events and therefore contribute to the OOA fraction?

Reply: Thanks for the reviewer's question. As reported by a previous study by Wang et al. (2020), the fossil fuel organic aerosols (FFOA) could be oxidised to aqSOA under high RH conditions. Nevertheless, we thought that the scavenging of fog droplets also played a role in the decrease of BBOA and HOA. Because not all BBOA and HOA were water-soluble (Qiu et al., 2019), and the insoluble parts could be scavenged by fog droplets, resulting in the decrease of BBOA and HOA in the interstitial particular matter. The results of Collet et al. (2008) also showed that the biomass

burning tracers were scavenged efficiently by fog droplets, while the vehicle emission tracers showed a relatively lower scavenging efficiency. However, as the reviewer mentions, we could not rule out the transformation of POA to OOA through aqueous-phase reactions. We have added relevant discussions in the revised manuscript. (lines 526-530)

As shown in Fig. S9, the mass fractions of OOA increased, while the contribution of BBOA and HOA decreased from pre-fog periods to post-fog/foggy periods for the three fog events. Except for the scavenging of fog droplets, BBOA and HOA could also be oxidised to OOA through aqueous-phase reactions (Wang et al., 2021), thus resulting in the decrease contribution of BBOA and HOA.

Reference

Qiu, Y., Xie, Q., Wang, J., Xu, W., Li, L., Wang, Q., Zhao, J., Chen, Y., Chen, Y., Wu, Y., Du, W., Zhou, W., Lee, J., Zhao, C., Ge, X., Fu, P., Wang, Z., Worsnop, D. R., and Sun, Y.: Vertical Characterization and Source Apportionment of Water-Soluble Organic Aerosol with High-resolution Aerosol Mass Spectrometry in Beijing, China, ACS Earth and Space Chemistry, 3, 273-284, https://doi.org/10.1021/acsearthspacechem.8b00155, 2019.

Wang, J., Ye, J., Zhang, Q., Zhao, J., Wu, Y., Li, J., Liu, D., Li, W., Zhang, Y., Wu, C., Xie, C., Qin, Y., Lei, Y., Huang, X., Guo, J., Liu, P., Fu, P., Li, Y., Lee, H. C., Choi, H., Zhang, J., Liao, H., Chen, M., Sun, Y., Ge, X., Martin, S. T., and Jacob, D. J.: Aqueous production of secondary organic aerosol from fossil-fuel emissions in winter Beijing haze, P. Natl. Acad. Sci. USA, 118, https://doi.org/10.1073/pnas.2022179118, 2021.

Page 20 line 533: "which are still unclear." In the introduction the author describe pretty well the topography and meteorological conditions of the SCB. If what the author meant is that their effects, together with emission sources, on haze formation processes are unclear, then the author may want to rephrase this sentence accordingly.

Reply: Thanks for the reviewer's suggestion. We have rephrased this part. (lines 538-540)

The formation process of haze pollution in SCB might be different from those in NCP, YRD, and PRD due to the unique topography, meteorological conditions and emission sources, which are still unclear.

Page 20 lines 545-548": "Due to the limitation of the present study, the parameters which are indicative of the pathways of nitrate formation are not characterised. The major precursors contributing to a large amount of OOA are not clear yet. In addition, how controlling BBOA will affect the atmospheric visibility, radiative forcing, and climate change in SCB needs further investigation." I would rephrase these sentences. The author emphasize the need to further investigate how controlling BBOA will affect radiative forcing etc.. while a better understanding of the gas/particles emitted by biomass burning (and not solely BBOA) and how they will contribute/affect the Haze formation processes, may help to implement appropriate POA emission control to reduce the occurrence of haze.

Reply: Thanks for the reviewer's suggestion. Indeed, our description in this part might not be comprehensive. We have rephrased this part as the reviewer suggests. (lines 552-560)

Due to the limitation of the present study, the parameters which are indicative of the pathways of nitrate formation are not characterised. The major precursors contributing to a large amount of OOA are not clear yet. In addition, further investigation is needed to gain a better understanding of the gas/particles emitted by biomass burning or other primary emissions and how they will affect the haze formation processes, which may help to implement appropriate POA emission control to reduce the occurrence of haze. Nonetheless, the results in this study implied that controlling primary emissions (such as biomass burning and vehicle exhaust) and precursors of secondary aerosols (e.g., $NOx$, $SO_2$, and VOCs) during severe haze periods will benefit the improvement of air quality in SCB.

Page 20 line 549: "In spite of the deficiencies". I would simply use "Nonetheless," as the author already stated the limitation of the approach beforehand.

Reply: Thanks for the reviewer's suggestion. We have replaced 'In spite of the deficiencies' with 'Nonetheless' in the revised manuscript. (lines xxx)

Nonetheless, the results in this study implied that controlling primary emissions (such as biomass burning and vehicle exhaust) and precursors of secondary aerosols (e.g., $NOx$, $SO_2$, and VOCs) during severe haze periods will benefit the improvement of air quality in SCB.

Page 27 – line 609 - Figure 6 "of different OA compositions"     OA sources or OA factors

Reply: Corrected. (line 618)

[Figure]

Fig. 6 Average mass fraction of different OOA, BBOA, and HOA (a) in OA and (b) as a function of OA mass concentration. The diurnal variation of different OA factors compositions and their mass contributions are shown in (c) and (d).

Page 28 – Figure 7 and caption: why using "lg(ALWC)" instead of ALWC?

Reply: The ALWC simulated by ISORRIPIA increased dramatically for the conditions with RH > 95 %. The ALWC simulated for the conditions with RH > 95 % are higher 2-3 orders of those with RH ranging from 60-90 %. It is not obvious to observe the effect of ALWC on OOA if mapping the data points with ALWC directly, so we used the logarithm of ALWC to map the data points, which could show the variation of OOA with ALWC clearer.

Page 30 - Figure 9: Have height axis in meter might be easier for the reader

Reply: Thanks for the reviewer's suggestion. We have changed the height axis to height above ground level (AGL) in Figure 9.

[Figure]

Fig. 9 Simulation results of 48 h backward air parcels cluster-mean trajectories during the campaign. The lines in black, blue, red, and purple represent the mean trajectories of Cluster1 to Cluster4, respectively. The pie charts show the average mass contribution of different chemical compositions to $PM_{2.5}$ for each cluster. The lower panel shows the height profile (above ground level, AGL) for different air parcels clusters along their transport paths.